# Aseismic slip and recent ruptures of persistent asperities along the Alaska-Aleutian subduction zone

Bin Zhao [1,2✉], Roland Bürgmann [3✉], Dongzhen Wang [3], Jian Zhang [1,4], Jiansheng Yu [1] & Qi Li [1]

The frictional properties and slip behaviors of subduction thrusts play a key role in seismic and tsunami hazard assessment, especially in weakly coupled "seismic gaps". Here, we rely on GPS observations in the Shumagin Gap of the Aleutian subduction zone to derive the slip distribution of the 2020 Mw 7.8 Simeonof Island, Alaska earthquake and of the subsequent afterslip during the first 87-day period. Our modeling results show that the mainshock ruptured at depths of ~30–40 km beneath Simeonof Island. Kinematic and stress-driven models indicate that the afterslip occurred both updip and downdip of the mainshock rupture. Physically plausible locking models derived from interseismic GPS velocities suggest that the 2020 Simeonof and 2021 Mw 8.2 Chignik earthquakes ruptured persistent asperities on the subduction thrust. We infer that there are several additional persistent asperities at depths of 20–50 km west ~157°W. However, it is still uncertain whether there are additional locked asperities at shallow depths because of the current lack of geodetic observations close to the trench.

[1] Key Laboratory of Earthquake Geodesy, Institute of Seismology, China Earthquake Administration, Wuhan 430071, China. [2] Institute of Disaster Prevention, Yanjiao, Sanhe City, Hebei Province 065201, China. [3] Department of Earth and Planetary Science and Berkeley Seismology Lab, University of California, Berkeley, CA 94708, USA. [4] Present address: Mengcheng National Geophysical Observatory, School of Earth and Space Sciences, University of Science and Technology of China, Hefei, China. ✉email: zhaobin@cgps.ac.cn; burgmann@seismo.berkeley.edu

Subduction zones are home to a substantial fraction of the seismic potential of the Earth, releasing ~90% of the total global seismic moment[1]. In the last two decades, there were several Mw > 8.5 megathrust earthquake ruptures in northeast Japan, Sumatra, and the Chile subduction zones, some of which caused large tsunamis. Recent advances in geodetic and geophysical observations have illustrated a wide spectrum of slip behaviors at different depths of subduction zones, including regular earthquakes, low frequency earthquakes (LFEs), very low frequency earthquakes (VLFEs), episodic tremor, and slow slip (ETS), and aseismic slip events and afterslip[2]. A wide spectrum of fault slip behaviors has also been documented in laboratory friction experiments on rock samples and in numerical simulations[3–6]. On the basis of these observations, a comprehensive description of subduction zones involves interseismically "fully locked" seismic asperities (from 5–10 km to 30–70 km in depth) that are bounded by aseismic and conditionally stable zones at a shallow depth that may host slow tsunami earthquakes, and by conditionally stable areas and aseismic sliding zones hosting smaller seismic asperities at greater depth[7,8]. In addition to the first-order, depth-dependent seismic characteristics, subduction zones also exhibit heterogeneous plate coupling along strike. Some segments of subduction zones are strongly coupled during the interseismic period and release accumulated elastic strain energy through fast earthquake ruptures. Variations in the megathrust faulting mechanism and seismic potential are probably related to the geological structure of the forearc and/or incoming oceanic plate, slab age and buoyancy, width of the seismogenic zone, and megathrust curvature[9–13].

Some subduction zone segments apparently creep and release a large fraction of their slip budget aseismically[14]. Some of these were identified as "seismic gaps" based on their lack of large historic earthquakes. Given the limited records, it is often uncertain whether more weakly-coupled "seismic gaps" have the ability to produce a great megathrust earthquake and generate large tsunami. For example, the northeast Japan subduction zone was generally not thought to be capable of generating M > 8 earthquakes based on its partially coupled kinematics and earthquake history, until the 2011 Mw = 9.0 Tohoku-oki earthquake taught us otherwise[15]. Therefore, it is very important to better understand the slip behaviors during earthquake cycles and the spatial variation of frictional properties of the subduction interface, especially for the weakly coupled segments.

The Shumagin Gap of the Alaska-Aleutian subduction zone has been identified as a seismic gap with low coupling ratio. The ~200 km-wide gap is bounded by the rupture zone of the 1938 Mw 8.3 earthquake to the east[16] and by the 1946 Ms 7.4 but Mw 8.6 tsunami earthquake to the west[17], between which no great earthquakes have occurred in the last century (Fig. 1). Davies[18] and Sykes et al.[19] inferred that earthquakes in 1788 and 1847–1848 ruptured through the Shumagin Gap and triggered large tsunamis. Bécel et al.[20] proposed that the shallow plate boundary and splay faults in the overriding plate, as imaged by seismic reflection data in this gap, may have participated in the failure of the 1788 tsunamigenic earthquake. However, Witter et al.[21] pointed out that there is no geologic evidence of great (M > 8) earthquakes and tsunamis in this seismic gap in the last ~3400 years, and they questioned the existence of the previously inferred great ruptures which were mainly based on secondhand Russian accounts. Geodetic observations suggest that the coupling ratio of this section is very low compared to that along the 1938 rupture zone to the east, where the subducting plate appears to be strongly coupled with the overriding plate[22–24]. Although the Shumagin Gap lacks well-documented great earthquakes, there were several medium-sized historical events located at depths of 30–50 km since the 1900s, including the 1917 Ms 7.4, 1948 Ms

7.5, and 1993 Ms 6.9 earthquakes. These earthquakes likely ruptured relatively small areas and did not generate notable tsunami.

On July 22, 2020, a moment magnitude Mw = 7.8 earthquake struck the Simeonof Island region of the Alaska-Aleutian subduction zone in the Shumagin Gap (Fig. 1). A tsunami warning was initiated after the earthquake and eventually canceled. The focal mechanism determined by the Global Centroid Moment Tensor (GCMT) indicates that the earthquake occurred on a shallow thrust at a depth of 36.8 km, consistent with a rupture of the well-determined slab interface. After 87 days, the October 19, 2020 Mw 7.6 Sand Point earthquake ruptured on a nearly NS-oriented fault plane with a strike-slip faulting mechanism to the west of the main event, triggered by the mainshock and subsequent afterslip[25]. One year later, the July 29, 2021 Mw 8.2 Chignik megathrust earthquake ruptured the subduction thrust just east of the 2020 Simeonof rupture zone at a similar range of depths as the 2020 event[26] (Fig. 1).

In this work, we demonstrate the coseismic slip distribution and subsequent afterslip distribution inverted from onshore Global Positioning System (GPS) data associated with the 2020 Simeonof earthquake. We do not consider contributions from viscoelastic relaxation of the oceanic mantle and mantle wedge, since they are usually insignificant compared to afterslip during the early postseismic period of M < 8 events[27]. Different from the results presented by Crowell & Melgar[28], we suggest that afterslip occurs on both the updip and downdip sides of the coseismic rupture when analyzing GPS data during the first 87-day period. Additionally, we construct a physically reasonable interseismic asperity model, taking into account the location of large historic and recent megathrust ruptures, the 2020 Simeonof and 2021 Chignik earthquakes, which gives a different spatial distribution of coupling and slip compared to previous models based solely on kinematic inversions of GPS data. Finally, informed by the recent activity and previous historic ruptures, we attempt to interpret the frictional properties and slip behaviors during earthquake cycles in the Alaska-Aleutian subduction zone.

## Results and discussion

**Coseismic slip distribution**. The coseismic slip distribution depends on the chosen weight of the smoothing constraint. There is a clear tradeoff between the model roughness and weighted residual sum of squares (WRSS), depending on the smoothing weight. We determine the optimal smoothing weight $\beta$ of 0.0012 on the basis of visual inspection of the L-curve (Supplementary Fig. 1). The preferred coseismic slip distribution with rake fixed at 90° is shown in Fig. 2 (see "Methods"). The earthquake rupture is located in the eastern part of the Shumagin Gap, and the coseismic slip concentrates between 30 and 40 km depth beneath the Shumagin Islands with a peak slip of 2.6 m, and it does not reach the trench. The checkerboard tests (see "Methods") indicate that the onshore GPS data can roughly constrain the depth range of coseismic slip (Supplementary Fig. 2). Considering a range of smoothing parameters shown in Supplementary Fig. 1 and assuming a rigidity of 50 GPa, the moment released by this event is a 6.9–7.4 × $10^{20}$ $Nm$, equivalent to Mw 7.82–7.84. Tests allowing for a strike-slip component in the coseismic models and with a different regularization method show similar slip distributions, although the peak slip magnitude and location changes somewhat (Supplementary Fig. 3).

**Kinematic afterslip distribution**. Similar to the coseismic modeling, the inverted afterslip distribution is sensitive to the chosen smoothing factor. In Fig. 3a, we display the cumulative afterslip distribution with a representative smoothing factor $\beta$ of 0.02. It

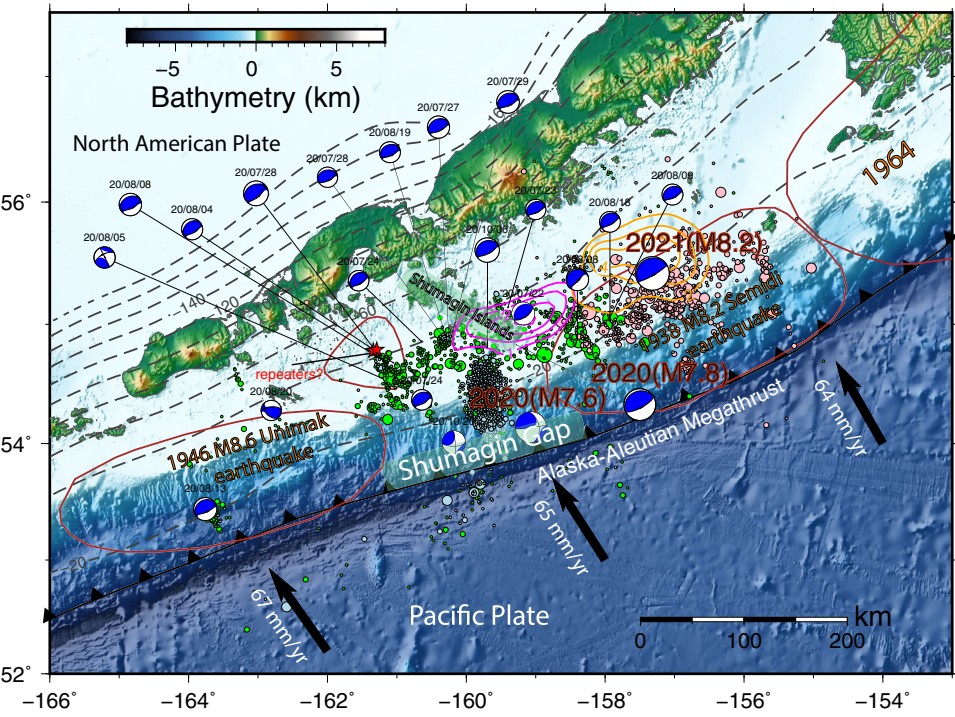

**Fig. 1 Seismotectonic setting of the Alaska-Aleutian megathrust.** The coseismic slip pattern for the 2020 Simeonof (Fig. 2) and 2021 Chignik earthquakes from U. S. Geological Survey (USGS) (https://earthquake.usgs.gov/earthquakes/eventpage/ak0219neiszm/finite-fault, last accessed August 22, 2021) are outlined in magenta and orange contours, respectively. The beach balls show focal mechanisms for M ≥ 5 earthquakes from the Global Centroid Moment Tensor catalog (GCMT) since the occurrence of the 2020 Simeonof mainshock. Aftershocks of the 2020 Simeonof earthquake until the 2020 Mw 7.6 Sand Point earthquake, 30-day aftershocks after the Sand Point event, and 20-day aftershocks following the 2021 earthquake from the USGS are shown with green, light blue, and pink circles, respectively, with size scaled by earthquake magnitude. Two red stars are similar earthquakes detected by Igarashi & Kato[62] that occurred in 1995 and 2011 and may indicate fault creep. Dashed black lines depict the depth contours from the Slab2 model[65]. Dark red curves delineate the approximate rupture areas of the three great historical earthquakes based on the distribution of aftershocks. Black vectors show the Pacific Plate velocities with respect to the North American Plate[74].

shows that the inferred afterslip appears to surround the coseismic rupture zone, especially in the western part. The inversion suggests very little afterslip to the east of the rupture; i.e., the rupture zone of the subsequent 2021 Chignik earthquake[26], and more substantial slip in the adjoining portion of the Shumagin Gap segment to the west. We find that the afterslip models with zero-slip constraint based on the various coseismic models[28–31] produce similar data fits to the GPS observations, and are mostly insensitive to the chosen rake constraint (Supplementary Fig. 4). If the zero-slip constraint in the coseismic zone is not applied, the afterslip also appears on the periphery of the coseismic rupture with the majority of afterslip in the western part when using the zeroth-order Tikhonov regularization approach (Supplementary Fig. 5). Taking into account a range of smoothing factors, the moment release during the first 87-day period is estimated to be $2.87$–$3.09 \times 10^{20}$ Nm, equivalent to an Mw 7.57–7.59 earthquake. This is approximately 38–44% of the moment released by the coseismic rupture. The observed cumulative postseismic displacements are fit well by the kinematic afterslip model in Fig. 3a except for the vertical observations close to volcanos on Unimak Island (Fig. 1), yielding a weighted root-mean-square (WRMS) value of 2.2 mm.

**Stress-driven afterslip distribution**. For comparison with the kinematic afterslip inversions, we consider stress-driven forward models under the three scenarios of downdip-only, updip-only, and fully surrounding the coseismic slip zone (see "Methods"). Although we have explored forward simulations over a wide range of model parameters, the optimal constitutive parameters that

minimize the misfit between the observed (see "Methods") and modeled GPS time series yield a relatively large normalized $\chi^2$ value of 29.7 in the first scenario (with a WRMS of 12.0 mm for the cumulative displacements). The determined frictional parameter $(a - b)\sigma_n = 1.83$ MPa and the reference velocity $v_0 = 4.99$ m/yr (Supplementary Fig. 6a). In this case, the aseismic afterslip is only allowed on the downdip extension of the modeled coseismic slip zone and has a peak slip of 1.4 m. This downdip afterslip model underestimates the surface motion at several stations close to the trench and predicts a large discrepancy in displacement direction (Supplementary Fig. 6b). Extending the upper bounds for the constitutive parameters in the Bayesian inversion process does not decrease the normalized $\chi^2$ value, indicating that the pure downdip afterslip cannot reproduce the postseismic observations.

In contrast to the first scenario, the second scenario represents the end-member case of an afterslip model with only updip afterslip. The best-fitting parameters $(a - b)\sigma_n = 0.59$ MPa and $v_0 = 0.60$ m/yr yield a normalized $\chi^2$ value of 9.3, much smaller than that in the first scenario (Supplementary Fig. 7a). The fit (with a WRMS = 5.7 mm) between the observed and predicted postseismic transients is shown in Supplementary Fig. 7b. This model produces a better fit to observations at AC28 and AC12, but underpredicts the postseismic displacements at AB07 and AC21.

As for the third scenario, the optimal stress-driven afterslip model produces a normalized $\chi^2$ value of 3.0. In this case, afterslip occurs around the full periphery of the coseismic rupture zone with a peak value of ~1.1 m. The optimal frictional parameter and reference velocity are 0.42 MPa and 0.16 m/yr, respectively

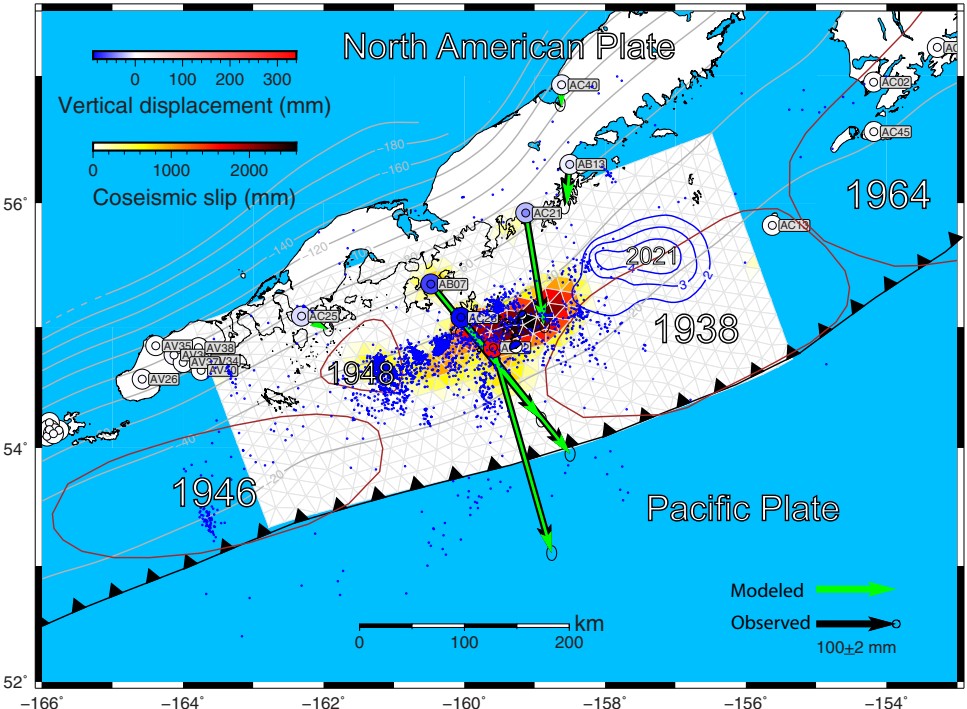

**Fig. 2 Comparison of observed and predicted 3-D coseismic displacements associated with the 2020 Mw 7.8 Simeonof Island, Alaska earthquake.** The error ellipses represent 95% confidence. The large and small colored circles represent the observed and predicted vertical displacements, respectively. Solid gray lines depict the depth contours of plate interface from the Slab2 model[65]. Small blue dots are aftershocks from the U.S. Geological Survey (USGS) catalog during the 87 days up to the Mw 7.6 Sand Point aftershock. Brown curves denote the rupture areas of the 1938, 1946, 1948, and 1964 earthquakes. The 2021 Mw 8.2 Chignik earthquake coseismic slip pattern from USGS is outlined in blue contours with slip ≥2 m.

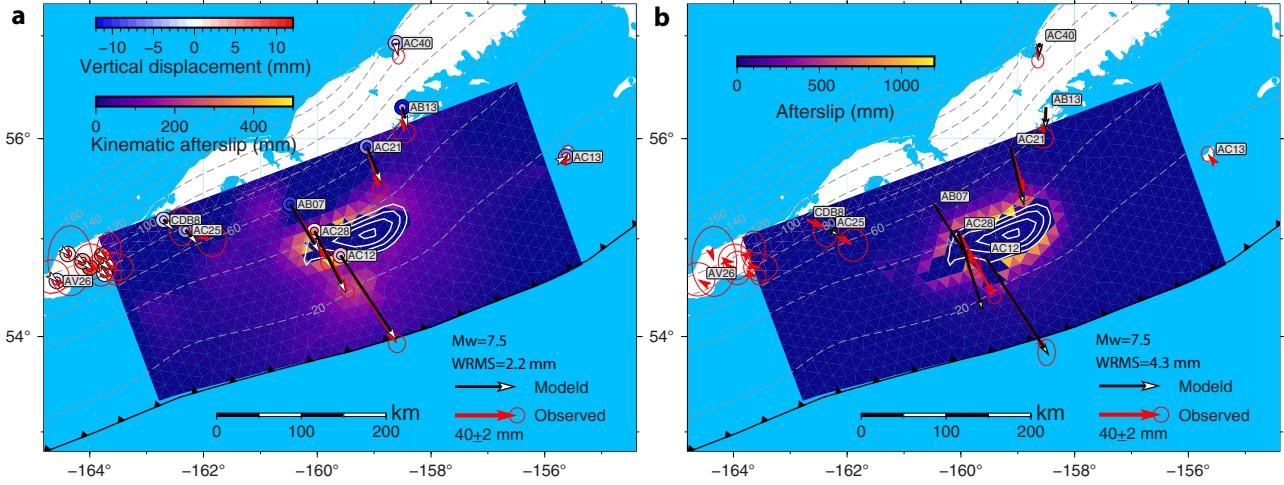

**Fig. 3 Kinematic afterslip model and stress-driven afterslip model. a** Cumulative afterslip distribution in the first 87 days inverted from Global Positioning System (GPS) postseismic transients. The weighted root-mean-square (WRMS) misfit for this kinematic model is 2.2 mm. White lines represent the contours of coseismic slip distribution of the 2020 Simeonof earthquake. Afterslip is not allowed within the ~1.0 m slip contour of the coseismic rupture. Dashed gray lines depict the depth contours from the Slab2 model[65]. **b** Stress-driven, frictional afterslip in the first 87 days (WRMS = 4.3 mm) with the optimal constitutive parameters based on the third scenario (afterslip on velocity-strengthening megathrust surrounding the coseismic rupture). Note the different color scales in the stress-driven vs. kinematic afterslip models.

(Supplementary Fig. 8). Figure 3b shows the stress-driven afterslip distribution and a comparison of the observed cumulative postseismic displacements with the model predictions. The GPS observed cumulative postseismic transients can be fit reasonably well by the optimal stress-driven afterslip model, with a WRMS of 4.3 mm, compared to 2.2 mm for the optimal kinematic afterslip model. The position time series from this afterslip model are generally in agreement with the observations

(Fig. 4). Frictional afterslip models, which are based on other coseismic models[29,31], all yield relatively large normalized $\chi^2$ values (4.2–4.5, Supplementary Figs. 9–11) compared to our preferred model (normalized $\chi^2$ of 3.0). Models invoking a coseismic strike-slip component also cannot improve the data fit (Supplementary Figs. 9–11). We could probably improve the fit to the data by considering spatial variations in frictional parameters as explored in Johnson et al.[32] for the afterslip following the 2004

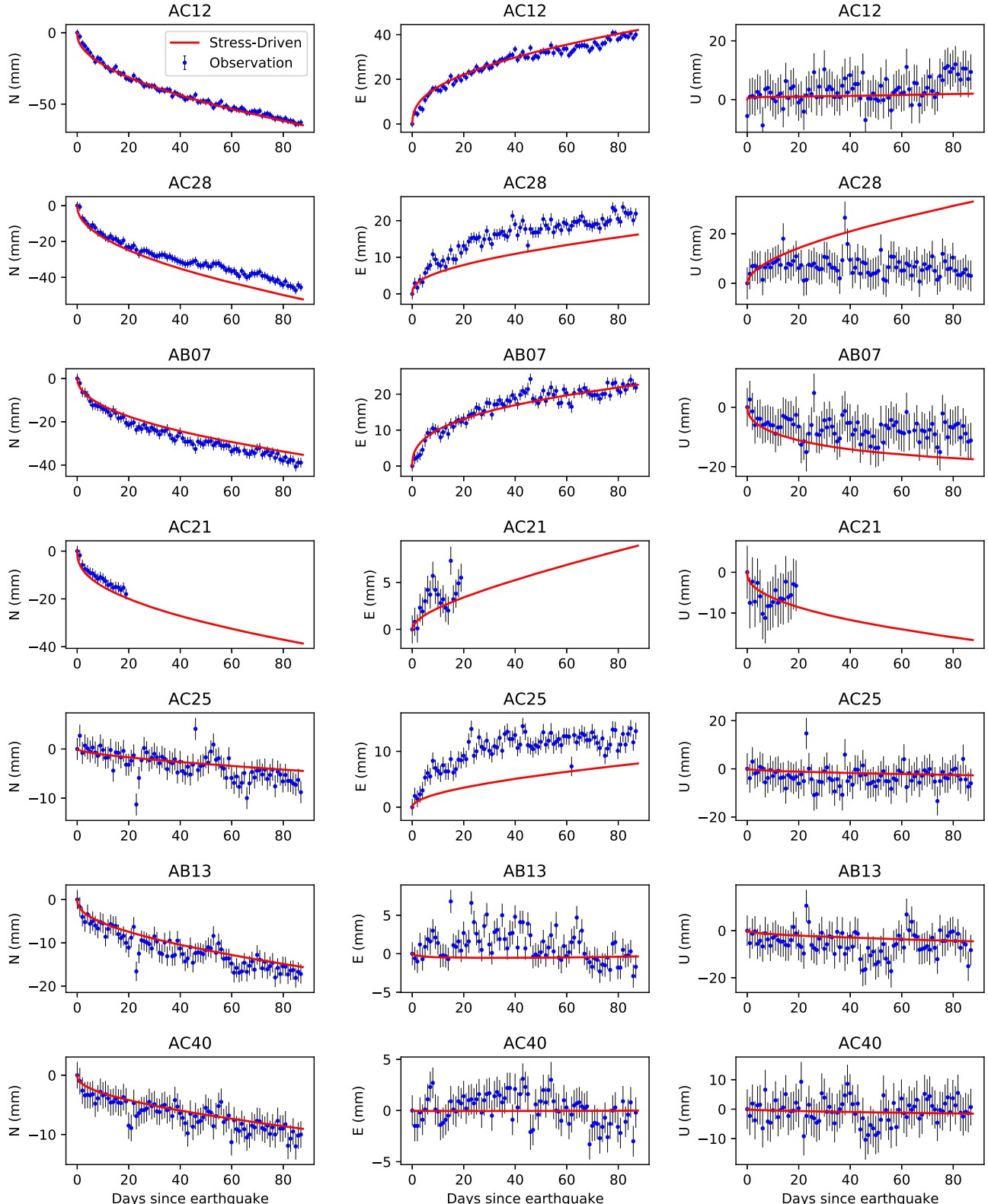

**Fig. 4 Comparison of the Global Positioning System (GPS) observed and modeled postseismic position timeseries.** Blue dots with error bars are observations and solid red lines represent predictions from the stress-driven afterslip model in the third scenario in the first 87-day period.

Mw 6.0 Parkfield earthquake. However, we do not think we can justify doing so given the small number of GPS sites. The moment released by our preferred stress-driven afterslip is $2.2 \times 10^{20}$ Nm, equivalent to an earthquake of Mw 7.5. The afterslip-to-coseismic

moment fraction is 31%, which is smaller than that of the kinematic afterslip model. This model does not predict the substantial afterslip found in the kinematic inversion in an area of abundant aftershocks to the west of the rupture (Fig. 3a).

**Interseismic asperity modeling.** First, we consider an end-member scenario in which the subduction thrust is locked along the full extent of historic ruptures, which was mainly determined from the distribution of aftershocks (see "Methods", Supplementary Fig. 12). The inferred rupture zones for the 1938 Semidi earthquake and the 1964 Prince William Sound earthquake are from Benz et al.[33], and the rupture zone of the 1948 Ms 7.5 earthquake is based on Boyd et al.[34]. For the 1946 Unimak tsunami earthquake, we here use the rupture zone determined by López & Okal[17]. We do not take the 1917 Ms 7.4 earthquake explicitly into account, since it ruptured the same zone of the 2020 Simeonof earthquake, possibly to a smaller extent[30]. This model overestimates the magnitude of the observed velocities at all GPS stations, especially in the western part of the modeled region (Supplementary Fig. 12). As expected, it yields a high WRMS value of 7.1 mm/yr and an extremely large normalized $\chi^2$ of 179 (see "Methods"). This modeling result implies that the size of persistent asperities must be much smaller than the rupture extent determined from historical aftershock zones.

Next, we examine whether interseismic asperity models, in which only core (high coseismic slip) zones of historical earthquakes are fully locked, can fit the GPS observations. We define the backslip asperities for the 1964 Alaska earthquake based on the coseismic slip model from joint inversion of seismic, tsunami, and geodetic data by Ichinose et al.[35]. Our triangular mesh of the plate interface only covers the westernmost asperity of the 1964 Alaska earthquake. As for the 1938 Mw 8.2 earthquake, we consider two distinct asperities, a large asperity in the east and a smaller one in the west, according to a simple coseismic slip model inferred from tsunami waveforms by Johnson & Satake[16] and to forward simulations of tsunami propagation for several scenario coseismic slip distributions by Freymueller[36]. Because there are no available detailed slip models for the 1946 Unimak tsunami and 1948 Ms 7.5 earthquakes, we treat their sizes and locations by trial and error. The modeling result shows that it underestimates the GPS observations in the Shumagin Gap (Supplementary Fig. 13), suggesting additional persistent asperities should be invoked.

Finally, we determine the preferred backslip rate distribution by trial and error, relying on available coseismic slip models of historical earthquakes and of the most recent ruptures. The asperity models that include the additional persistent asperities corresponding to the 2020 Simeonof and 2021 Chignik earthquakes significantly improve the data fit. Figure 5 shows a plausible backslip rate (coupling) distribution that well fits the observed interseismic velocities, yielding a normalized $\chi^2$ of 13 (WRMS = 1.9 mm/yr). In this scenario, the size of the western asperity of the 1938 earthquake is further reduced compared to that in Supplementary Fig. 13. The relatively poor fit to the displacement azimuths of stations above the 1938 and 1964 rupture zones (Supplementary Fig. 14) is possibly caused by inaccurate asperities inferred from coarse slip models[16,35,36], may indicate additional heterogeneity in along-arc coupling, or may be related to more complex forearc deformation[37] than assumed in the correction applied by Li and Freymueller[22]. Although there is little overlap between the inferred core asperities of the 1938 and 2021 earthquakes, it appears that the dynamically ruptured zone of the 2021 earthquake may have re-ruptured deeper parts of the 1938 rupture zone (Fig. 5). According to the inferred interseismic coupling, the maximum slip deficit on the fully locked sections of the subduction thrust has reached 5.4 m, during the 83 years since the 1938 earthquake.

We note that previously suggested[22–24] shallower and larger zones of coupling can fit the measured interseismic velocities on the Shumagin Islands equally well (Supplementary Fig. 15). However, this scenario fails to explain the rupture locations of the 2020 Simeonof and 2021 Chignik events, and the existence of large-sized shallow asperities to the west of the 1938 rupture zone seems unreasonable[16,36]. Another possible scenario, invoking a second small asperity updip of the 2020 rupture zone (above 20 km), can also fit the GPS observations equally well (Supplementary Fig. 16). In this case, the sizes of the two asperities are smaller than those in Fig. 5 and Supplementary Fig. 15. We also note that the location of the asperity for the 1946 Unimak tsunami earthquake is not well determined. If this asperity is excluded, the misfit between the observed and predicted velocities decreases slightly (Supplementary Fig. 17), and it is possible that the actual rupture asperity of this event lies further west in an area lacking GPS coverage.

Supplementary Figure 18 shows a plausible backslip rate (coupling) distribution with the least complexity and only five core asperities. This simple model can also achieve a reasonable fit since contributions due to the small and shallow asperities corresponding to the 1946 Unimak tsunami earthquake and to the western rupture of the 1938 earthquake are very small compared to other large asperities (Supplementary Fig. 19). These tests suggest that the obtained characterizations of coupling models in Fig. 5 and Supplementary Figs. 15–18 are consistent but not unique representations of the slip-deficit distribution.

Overall, the inferred locked asperities are only roughly 25% of the size of the rupture areas determined by aftershock distributions or by source inversions from geodetic and teleseismic waveforms, similar to findings by Bürgmann et al.[9] and Johnson et al.[38] off Kamchatka and NE Japan, respectively. The size of the asperity areas decreases from NE to SW along strike, and the extent of conditionally stable and creeping areas increases correspondingly. This spatial pattern of asperity sizes is consistent with the previously inferred first-order, along-strike variation in coupling ratio[22–24].

**Comparison with previous coseismic and postseismic models.** To first order, our preferred coseismic rupture zone overlaps with the other coseismic slip models constrained by varying datasets[28–31], showing that the coseismic slip dominantly occurs on the plate interface at depths from 30 to 40 km and does not reach the trench (Supplementary Fig. 20). However, differences are also apparent. The peak slip area of our model is located further east compared to other models. In addition, our model does not have distinct secondary asperities. We find that the kinematic afterslip models with zero-slip constraint based on the other coseismic models produce similar fit to GPS observations compared to models with our preferred coseismic model (Supplementary Fig. 4).

While our coseismic slip distribution is generally in agreement with that of Crowell & Melgar[28], differences exist between our kinematic and stress-driven afterslip models and theirs. The kinematic afterslip model of Crowell & Melgar[28] features a majority of afterslip downdip of the coseismic rupture, although they indicate that the existence of updip afterslip cannot be ruled out. In contrast, our inferred afterslip model from the kinematic inversion with zero-slip constraint suggests afterslip surrounding the coseismic peak slip zone (Fig. 3a). We also find a substantial afterslip to the west of the coseismic rupture in an area of widespread aftershock activity, but not to the east, where the subsequent Chignik rupture occurred. Kinematic afterslip inversions without zero-slip constraint and using the same triangular mesh and rake constraint as used in Crowell & Melgar[28] also show significant updip afterslip, as well as downdip afterslip in the west of the coseismic rupture area (Supplementary Fig. 5). We suspect the differences can be attributed to the method of calculating the Green's functions. Different from the triangular

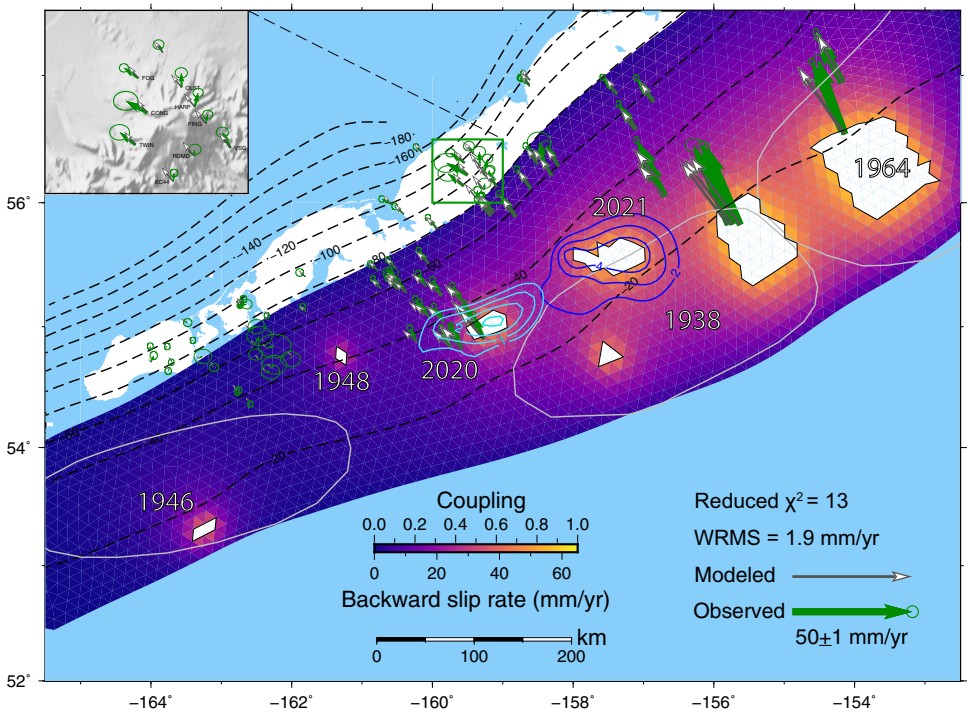

**Fig. 5 The preferred interseismic backward slip rate (coupling) distribution in the Alaska subduction zone based on historical and recent rupture zones.** Shown in white are locked asperities based on the location of past ruptures with a backward slip rate of 65 mm/yr, equals to the relative plate convergence rate between the Pacific plate and North American plate[74]. Note that the 1964 asperity represents only the westernmost of three coseismic slip patches of the Mw 9.1 event[35]. Colors of triangular elements denote backward slip rate (coupling) calculated from the forward Boundary Element Method (BEM) model. Green and gray arrows show the observed and predicted station velocities. Dashed lines depict the depth contours from the Slab2 model[65]. Gray outlines labeled with year are the rupture zones of large historical earthquakes. Light blue and deep blue lines represent the contours (in meters) of the coseismic slip distribution of the 2020 Simeonof and 2021 Chignik earthquakes, respectively. Inset shows the observed and modeled Global Positioning System (GPS) data around Veniaminof volcano (green rectangle).

dislocation method we used, Crowell & Melgar[28] employed a point-source representation.

Our modeling results from three frictional afterslip scenarios indicate that neither the pure downdip afterslip model nor updip-only afterslip can reproduce the observed GPS time series and that stress-driven models with afterslip surrounding the coseismic rupture can better fit the observations. The frictional afterslip models produce less slip to the west of the rupture than found in the kinematic inversion. That is, the coseismic stress change cannot be solely responsible for the observed slow slip in that area. This may suggest the occurrence of a triggered slow slip event, which released a previously built-up slip deficit on that portion of the Shumagin Gap[39]. A similar pattern of updip and downdip afterslip distribution also occurred in the aftermath of the 2005 Mw 8.7 Nias-Simeulue megathrust earthquake, the 2007 Mw 8.4 Bengkulu earthquake, and the 2008 Mw 7.2 North Pagai earthquake along the Sunda subduction zone and the 2016 Mw 7.8 Pedernales earthquake along the central Ecuador subduction zone[40–43]. Compared to the stress-driven afterslip model in Fig. 3b, the distributed nature and low magnitude of kinematic afterslip (Fig. 3a) is mostly due to the small number of GPS stations used in the kinematic inversion.

**The relationship between afterslip and aftershocks.** One mechanism of aftershock generation is the unstable failure of small asperities loaded by aseismic slip, supported by evidence that both aftershocks and afterslip occur in the area surrounding the peak coseismic slip zone and follow a similar temporal decay[44–46]. We compare the spatial distribution and temporal

evolution of our favored stress-driven afterslip model with the aftershocks from the U. S. Geological Survey catalog (USGS). We admit that this original unrelocated aftershock distribution comes with large uncertainties, but it still provides the first-order pattern and temporal evolution. Figure 2 shows that the aftershocks during the first 87 days are generally distributed at the margin of the coseismic rupture zone, especially along the western edge. We note that this western aftershock zone overlaps with the zone of a possible triggered slow-slip episode discussed in the previous section. The existence of aftershocks within the rupture zone may be interpreted as a result of frictional heterogeneity[47] or triggered events in the adjoining crustal blocks. It is notable that there are relatively few aftershocks in the inferred shallow afterslip zone, except for in the eastern section. We suspect that the poor correlation between the spatial distribution of the aftershocks and afterslip is in part caused by large uncertainties in the aftershock locations, since teleseismic event locations can be off by 10 s of km due to Earth structure effects[48]. It may also imply rather homogeneous velocity strengthening frictional properties on the shallow portion of the weakly coupled (creeping) Shumagin Gap (Fig. 5). In contrast, a large number of aftershocks were located in the shallow afterslip zone of the 2005 Mw 8.7 Nias-Simeulue, Sumatra earthquake[40], which appears to have been well coupled during the interseismic period, albeit the coupling on the shallow portion was poorly determined[49]. Aftershocks following the 2021 Chignik earthquake mainly concentrate on the shallow updip area (Fig. 1) and were probably triggered by the collocated aseismic afterslip, more similar to what was observed following the Nias-Simeulue event[40]. The background seismicity in the shallower portion of the Shumagin Gap is low compared to that of the

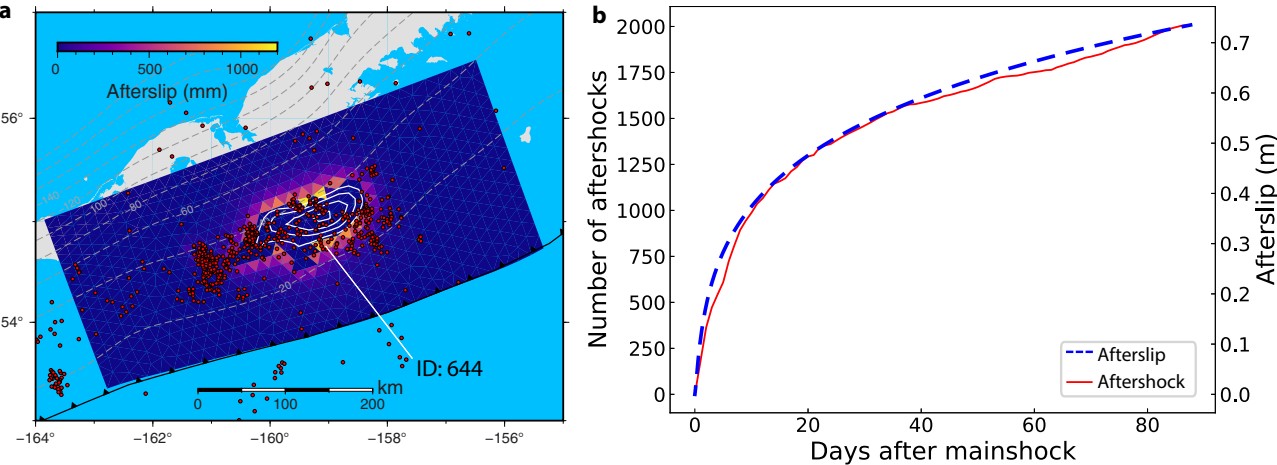

**Fig. 6 Spatiotemporal correlation between afterslip and aftershocks. a** Spatial distribution of afterslip and aftershocks in first 87 days after the 2020 Simeonof earthquake (red dots). **b** Comparison of the temporal decay of stress-driven afterslip with that of the aftershock activity. Red line shows the cumulative number of aftershocks near the slab interface near the coseismic rupture zone (Fig. 2). Dashed blue line represents postseismic afterslip evolution in the first 87-day period on a selected triangular element with index number 644 labeled in (**a**).

neighboring 1938 earthquake rupture zone[50]. This feature of low seismicity rates in some sections of creeping subduction zones also exists in the creeping subduction section in the northern Nazca-South American subduction zone[43].

Figure 6 shows a comparison of the cumulative number of Simeonof earthquake aftershocks together with the stress-driven afterslip evolution for the 87-day period leading up to the Sand Point event, as well as the spatial distribution of aftershocks and afterslip. We find that aftershocks and afterslip follow a similar temporal decay. A similar temporal evolution pattern of afterslip and aftershocks at timescales of days to years after an earthquake has been widely observed elsewhere after other well-documented earthquakes[40,44,51,52]. However, the aftershock-afterslip correlation may weaken over longer time scales (several years) due to an increasing contribution from viscoelastic relaxation to the crustal deformation and stress[52]. The good temporal but poor spatial correlation between afterslip and aftershocks probably reflects that many aftershocks were triggered by static and dynamic stress changes from the event, not just the failure of small asperities driven by surrounding afterslip. We also compare the seismic moment released by aftershocks to the moment release from the preferred stress-driven aseismic afterslip model (Supplementary Fig. 21). The local magnitudes of the aftershocks from the USGS catalog are converted to moment magnitude using an empirical relationship[53]. The seismic moment released by the aftershocks in the 87-day postseismic period is about two orders of magnitude smaller than the moment released by aseismic afterslip (Supplementary Fig. 21), suggesting that the coseismically increased stress surrounding the rupture is mainly relieved through aseismic slip in the first 87 days.

**Frictional properties**. Our distributed coseismic and afterslip models of the Simeonof Island earthquake illustrate that the frictional properties of the plate interface in the Shumagin Gap vary with depth. We argue for a transition from a velocity strengthening or conditionally stable region in the shallow portion (<~30 km) to velocity weakening at depths of 30–40 km, and then becoming velocity strengthening again at depths greater than ~40 km. Our preferred stress-driven afterslip simulation constrains a uniform frictional parameter $(a-b)\sigma_n$ of 0.42 MPa. This value is comparable to values estimated using similar models of afterslip for other earthquakes elsewhere[40,54]. Assuming the effective normal stress is 360 and 800 MPa at 18 and 40 km under

hydrostatic pore pressure conditions, respectively, the $(a-b)$ value is about $1\times10^{-3}$–$6\times10^{-4}$, which is at least one order of magnitude smaller than typical laboratory values of $10^{-2}$ at temperatures corresponding to depths greater than 18 km[55]. This probably means that the effective normal stress on the afterslip zones may be significantly reduced due to elevated fluid pressure, which has been inferred to exist in other subduction zones[56,57]. The elevated fluid pressure may also be a plausible factor for a low degree of interseismic locking[56]. The weakly coupled Shumagin Gap inferred from our interseismic asperity model (Fig. 5) may also reflect such elevated pore pressure. Tomographic seismic imaging and determination of the ratio of compressional to shear wave velocities along the megathrust could provide a test of this scenario[58]. Additionally, although the GPS data used here do not allow for resolving large variations of frictional properties as in Wang & Bürgmann[59], it is likely that the deep, temperature-controlled rheology of the megathrust differs from that in the shallow sections, as well as the lateral contrast as expected[25].

**Interseismic coupling and implications for fault slip behavior**. Several interseismic coupling models of the Alaska subduction zone were determined in previous studies, based on model inversions of geodetic measurements that revealed significant along-strike variation in the coupling ratio. In these models, the average coupling ratio strongly decreases from the highly coupled Kodiak segment in the northeast to the Sanak segment in the southwest[22–24,37]. In the Shumagin Gap segment, the estimated coupling ratio drops from ~0.4 close to the trench to zero at a depth of ~60 km[22]. These depth-dependent coupling distributions inferred from kinematic inversion of GPS velocities seem physically implausible, since it is hard to explain the occurrence of the 2020 Mw 7.8 Simeonof and 2021 Mw 8.2 Chignik earthquakes in such a creeping dominated region. Recently, Xiao et al.[31] developed a new interseismic coupling model with the assumption that the inverted coupling follows a Gaussian distribution, rather than a decaying function, with depth. In this model the peak slip deficit is centered in the intermediate depth range instead of near the trench as found in previous models[22,24]. Although their new model does not fully agree with the asperity models presented in this study, it supports that the 2020 and 2021 events did not rupture on the poorly coupled megathrust interface. Our more physical interseismic asperity model, assuming full locking on core rupture asperities of historical and more

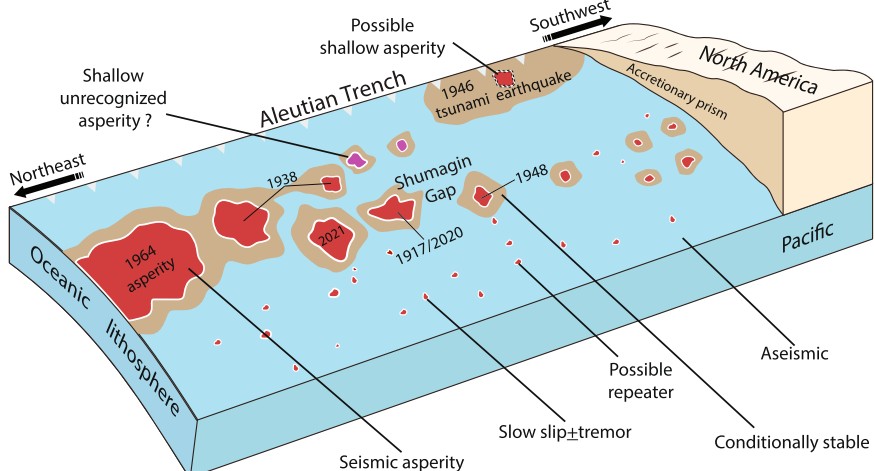

**Fig. 7 Summary schematic of the tectonic setting and slip behaviors of the plate interface along the Aleutian subduction zone.** The size of permanent core seismic asperities decreases from east to west and the size of conditionally stable and creeping areas increases correspondingly.

recent earthquakes, shows possible but not unique backslip scenarios that can fit the GPS measurements generally well, except in the areas above the 1938 and 1964 rupture zones as described above (Supplementary Fig. 14). Our preferred but non-optimized forward model in Fig. 5 produces a WRMS value of 1.9 mm/yr which is somewhat larger than 1.1 mm/yr for the inverted backward slip model in ref. [22] (see "Methods" for details on the GPS velocities used in misfit calculation). It is not surprising that they obtained lower residuals, since they also estimated the strike-slip component on each fault patch (Supplementary Fig. 14).

Our models are consistent with distinct and persistent asperities corresponding to the 2020 and 2021 earthquakes at depths of 20–40 km (Fig. 5). It is possible that there are additional smaller asperities at shallower depth (<~20 km), including a shallow and smaller western 1938 asperity (Supplementary Fig. 16). However, it is still uncertain whether there are additional asperities at shallower depth because of the lack of near-field geodetic observations close to the trench. We suggest that the better we know the size and location of asperities located at depths of ~30–50 km, the more confidently we can infer the state of coupling at shallower depths, since there are tradeoffs between asperities in the shallow and deeper depth intervals. Ultimately, seafloor geodetic observations in this and other partially coupled subduction zones will help us to better constrain the size and location of seismic asperities and to assess the associated tsunami earthquake hazard.

While details of the locking distribution are still not well resolved (Supplementary Figs. 13, 15–18), we can rule out scenarios with much larger locked zones, such as those associated with inferred historical rupture areas (Supplementary Fig. 12). No matter which interseismic asperity model we consider, our models indicate a different spatial distribution of coupling and slip compared to previous kinematic coupling models, and fully locked asperities make up only a fraction of the plate interface. Based on the interseismic coupling shown in Fig. 5 and assuming a shear modulus of 50 GPa, the moment deficit over the 1938 rupture zone bounded by the dark red curve in Fig. 5 that accumulated in the 83 years since 1938 is estimated to be $4.1 \times 10^{21}$ Nm (Mw = 8.3). After subtracting the moment released during the 2021 Chignik earthquake in the overlapping region with the 1938 rupture zone, the remaining cumulative moment deficit is estimated to be $2.9 \times 10^{21}$ Nm (Mw = 8.2). This suggests that there is still a substantial remaining earthquake potential along the section of the Alaska-Aleutian subduction zone that hosted the 1938 event.

We summarize the slip behaviors on the plate interface in the Alaska-Aleutian subduction zone during earthquake cycles in Fig. 7. While the inferred large-sized asperities corresponding to the 1938[36] and 1964[35] events are relatively shallow, we speculate that there are several unconnected smaller-sized persistent asperities at a depth of 20–50 km corresponding to moderate-sized earthquakes, such as the 1917 Ms 7.4, 1948 Ms 7.5 and 2020 Mw 7.8 Simeonof Island earthquakes in the Shumagin Gap, and the 2021 Mw 8.3 Chignik earthquake in the Semidi segment. In this paper, we focus on the spatial distribution of persistent locked asperities, that is we assume that patterns of interseismic coupling on the plate interface persist over time spans of one or more earthquake cycles. These asperities are fully locked during the interseismic period and are surrounded by partially coupled, conditionally stable zones and freely creeping areas. When the asperity builds up sufficient shear stress to overcome frictional resistance, the fault ruptures dynamically. However, the persistence of asperities does not necessarily produce characteristic seismic ruptures[60], and in some cases, multiple asperities may rupture together[61]. The rupture zone is not confined to the persistent, fully locked asperities, but extends into the surrounding conditionally stable zones, frequently producing moderate-sized earthquakes and failing to generate large tsunami. Subsequently, the stresses on the plate interface imparted by the coseismic rupture are relieved through aseismic slip in weakly coupled areas surrounding the rupture zone. The aseismic slip on the velocity strengthening zone will further trigger aftershocks or repeating earthquakes located near or contained within the afterslip zone[62]. As the afterslip rapidly decays with time, the viscoelastic relaxation of the oceanic mantle and the continental mantle wedge may eventually become the dominant deformation mechanism producing postseismic deformation and stress changes.

It is plausible, that ruptures occasionally break through the shallow portion and grow into infrequent great tsunami earthquakes facilitated by thermal pressurization and other weakening mechanisms at high slip speeds, as has been suggested for the 2011 Mw 9.0 Tohoku-oki event[4,5] and may have been the case for the 1946 Mw 8.6 tsunami earthquake in the western Shumagin Gap. Several characteristic structures on the shallow portion of the subduction zone may favor the occurrence of tsunami earthquakes, including active crustal-scale splay faults in the overriding plate, a small frontal prism, and a rough and thinly sedimented plate interface[20]. All these features have been identified in the Shumagin Gap as imaged in seismic reflection profiles by Bécel et al.[20] and by Shillington et al.[63].

## Methods

**GPS data processing**. The daily three-component GPS position time series used to study the coseismic and postseismic deformation associated with the 2020 Mw 7.8 Simeonof earthquake are from the Nevada Geodetic Laboratory (NGL) archive (http://geodesy.unr.edu)[64]. The average formal position uncertainties provided by NGL are less than half the observed scatter in position about the fitting function in Eq. (1). Thus, we simply rescale the formal position uncertainties by a factor of 2.5 to obtain more realistic uncertainties. In our study region, the original GPS position time series in the ITRF2014 reference framework contains a number of signals: a linear tectonic deformation rate, annual and semi-annual loading, offsets caused by equipment changes, coseismic offsets, including the 2018 Mw 7.9 Gulf of Alaska earthquake and the 2020 Simeonof event, and postseismic transients, as well as signals related to volcanic activity. To extract the coseismic and postseismic signals, we fit the 3-D position time series using a function of the following form:

$$y(t) = a + bt + c\sin(2\pi t) + d\cos(2\pi t) + e\sin(4\pi t) + f\cos(4\pi t)$$
$$+ \sum_{i=1}^{n} g_i H_i(t) + h\ln\left(1 + \frac{t}{\tau}\right) \qquad (1)$$

Where $a$ is the value at the initial epoch, $b$ represents a linear rate, the $sin$ and $cos$ functions represent annual and semi-annual signals, $H_i(t)$ denotes the Heaviside step function, $g_i$ represents the $ith$ jump of $n$ jumps due to coseismic and/or non-coseismic position changes, and the last term represents a logarithmic postseismic transient with a time constant $\tau$ and a postseismic coefficient $h$.

We solve for the parameters in Eq. (1) using a non-linear least squares method with bounds on the variables to extract the coseismic offsets directly and then isolate "pure" postseismic transients by removing non-postseismic signals from the raw position time series (Supplementary Fig. 22). The advantage of this method is that it can determine the time constant $\tau$ for each time series. We do not explicitly account for transient signals associated with volcanic activity, since the nearest actively-deforming volcano (Veniaminof volcano) is relatively far from the earthquake rupture region[24] and appears to have had little effects on the GPS sites we use during this time period.

**Coseismic slip model**. The focal mechanism of the 2020 Simeonof earthquake determined by the GCMT and aftershock distributions from USGS indicate that this event occurred on the plate interface. Thus, we directly define a realistic 3-D fault geometry based on the latest Slab 2.0 model[65] rather than constrain it using geodetic data. The modeled fault interface extends from the upper edge of the trench at 7 km depth based on its bathymetric expression to an average depth of ~80 km and has a length of approximately 500 km, covering a wide region for the determination of both the coseismic and afterslip distributions, but not for the interseismic asperity model. We discretize the slab surface into a mesh with 1180 triangular elements with an average side length of 15 km, which allows us to pursue a more complicated fault geometry than the commonly used rectangular patches.

We invert for the coseismic slip distribution from the GPS-derived 3-D coseismic offsets. The weighted inversion minimizes the L2 norm $||W(d - Gm)||_2 + ||\beta\nabla^2 m||_2$, where $m$ is the dip slip on each triangular dislocation, $G$ represents the displacement Green's functions, $d$ denotes the GPS data, and $W$ is the weight of the observations derived from their uncertainties. To avoid unreasonable oscillations of slip between adjacent triangular patches, a smoothing constraint is imposed using the Laplacian smoothing operator $\nabla^2$ and a smoothing factor $\beta$. We employed a scale-dependent umbrella operator to approximate the discrete Laplacian[66]. The Green's functions, which relate unit dip slip on each triangular mesh element to surface displacements, are computed using an improved triangular-dislocation algorithm in a homogeneous, isotropic, and elastic half-space[67]. We use the bounded variable least squares (BVLS) method[68] to impose slip bounds. We also test the performance of coseismic models with a strike-slip component (rake allowed to vary between 45°~135°) and with another regularization method, the zeroth-order Tikhonov (minimum norm) approach.

**Kinematic afterslip distribution**. We invert for distributed afterslip on the plate interface from the GPS-derived 3-D cumulative postseismic displacements in the first 87-day period and solve only for thrust slip in the dip direction on each triangular element, using a similar approach used in the coseismic slip inversion. However, here we employ the zeroth-order Tikhonov regularization method, since we find it is more robust than the one used in the coseismic slip inversion when the number of observations is limited and the displacement magnitude is small (Supplementary Fig. 5). Given that the afterslip usually occurs on the periphery of coseismic rupture zones, but there may be some overlap between afterslip and coseismic rupture zones with lower slip magnitude[60], we apply a constraint of zero slip in the region of peak coseismic slip (>1.0 m), which is determined by trial and error. We test the performance of kinematic afterslip models with the zero-slip constraint based on other published coseismic models and models allowing for a strike-slip component. We also examine afterslip distributions without the zero-slip constraint in the peak coseismic-slip zone.

**Stress-driven afterslip distribution**. The distributed afterslip inverted from postseismic displacements is, in essence, a purely kinematic model. To obtain physically plausible afterslip models, we conduct frictional afterslip modeling driven

by coseismic stress changes in a forward-modeling sense. We assume that afterslip is governed by a velocity-strengthening friction law, that is an empirically derived relationship based on laboratory rock friction experiments[44,54,69,70]. The law relates shear stress $\tau$ and normal stress $\sigma$ on the fault plane to the fault slip rate $v$.

$$\tau = \sigma_n\left[\mu + (a - b)\ln\left(\frac{v}{v_0}\right)\right] \qquad (2)$$

where $\mu$ is the nominal friction coefficient, $v_0$ is a reference slip rate, $(a - b)$ is a dimensionless frictional parameter, and $\sigma_n$ is the effective normal stress. For $(a - b) > 0$, an increase in sliding velocity will lead to an increase in the effective friction coefficient, and hence slip will not accelerate and occur aseismically. The expression of the velocity-strengthening law is:

$$v = 2v_0\sinh\left[\frac{\triangle\tau}{(a - b)\sigma_n}\right] \qquad (3)$$

where $\triangle\tau$ is the coseismic stress change. This formula assumes that the cumulative afterslip is much greater than the critical slip distance in rate-and-state friction laws and the coseismic stress change is significantly greater than the background stress[70].

The static coseismic stress changes on the fault plane are calculated based on the preferred coseismic slip model. We consider three different scenarios of afterslip distributions on the plate interface. In the first scenario, we assume that afterslip only occurs on the downdip side of the coseismic rupture zone, as suggested by Crowell & Melgar[28] and also similar to the afterslip distribution after the 2015 Mw 7.8 Gorkha earthquake[71]. The second scenario assumes that only the shallow updip extension of the coseismic rupture zone hosts aseismic slip following the event. This assumption of absence of downdip afterslip is physically unreasonable, since fault frictional properties generally become velocity strengthening at depths below the seismogenic zone and thus favor aseismic slip[60]. Finally, we assume that both the downdip and updip zones of the coseismic rupture accommodate stress-driven afterslip.

We determine the corresponding optimal constitutive parameters $v_0$ and $(a - b)\sigma_n$, which best fit the GPS observed time series of postseismic position using a Markov Chain Monte Carlo (MCMC) Bayesian sampler[72], for each scenario. We assume a uniform prior probability distribution for $v_0$ and $(a - b)\sigma_n$. Our Monte Carlo chain has 40,000 samples and produces 20,000 samples of the posterior distribution with a default thinning value of 1. The autocorrelation of the Markov chain decreases quickly with an increasing lag and becomes virtually zero within a lag of ~100–200 (Supplementary Fig. 23). We evaluate the goodness of fit between the observed and predicted position time series using the normalized chi-squared ($\chi^2$) misfit statistic expressed as

$$\chi^2 = \frac{1}{3N \times M}\sum_{i=1}^{3N}\sum_{j=1}^{M}[(\text{obs}_{ij} - \text{mod}_{ij})/\sigma_{ij}]^2 \qquad (4)$$

where $M$ is the number of epochs at each site and $N$ is the number of GPS stations. The $\text{obs}_{ij}$, $\text{mod}_{ij}$ and $\sigma_{ij}$ are the observed and predicted, and corresponding uncertainty of the 3-D position time series at the $i$th GPS station at epoch $j$.

We also test whether the stress-driven frictional afterslip models based on other coseismic models[29,31] and our own test model with a coseismic strike-slip component can improve the fit between the observed and modeled postseismic time series applying the same procedure as described above. We extend the length and width of the modeled fault planes in ref. [29] and ref. [31] to cover a wider region on which afterslip is allowed to occur.

**Checkerboard resolution tests**. We carry out spatial resolution tests to assess to what degree the inverted kinematic coseismic and afterslip distributions are well constrained by the onshore GPS data following the procedure in Bürgmann[9]. We first compute the predicted surface displacements at all stations where we have GPS data using a prescribed synthetic slip distribution with slip of 0 m or 1 m on $6 \times 5$ subfaults. Next, we apply 0–3 mm of random noise to the calculated displacements, then construct the variance-covariance matrix according to the observed uncertainties, and finally reinvert for the optimal slip distribution using the aforementioned approach. Supplementary Fig. 2 shows that we can resolve the first-order slip pattern of the deeper part of the modeled fault surface, if the noise applied to the predicted displacements is less than 3 mm. As expected, we cannot reproduce the slip pattern on the shallow subduction thrust close to the trench, even if not applying noise. Although the data has no ability to resolve the slip on the shallowest portion of the megathrust, it appears that we can still discern the slip pattern on the portion up to a few kilometers above the 20 km depth contour of the plate interface (Supplementary Fig. 2).

**Interseismic asperity modeling**. We employ an "asperity model" to infer a physically plausible distribution of persistent asperities. It is worth noting that the considered persistent asperities are based on the inference of overlap of earthquake rupture zones and interseismic slip deficit zones. In this simple model, the fully coupled patches are assumed to be confined to velocity-weakening asperities, which are fully locked during the interseismic period, and postseismic and interseismic creep occurs on the velocity strengthening area of the fault outside of the asperities[9,38]. This method relies on the knowledge of past ruptures. Fortunately, the historical earthquakes were relatively well documented and observations span a

substantial fraction of likely recurrence intervals. We use the Boundary Element Method (BEM) code POLY3D to solve for slip on the aseismic portion of the plate interface with stress boundary conditions loaded by back slipping of the fully locked asperities. This approach accounts for stress shadow effects in the vicinity and especially updip of the locked asperities[25,73]. We define the subduction interface from the latest Slab 2.0 model[65] similar with the mesh used for coseismic and afterslip modeling but covering a wider region (Fig. 5). The plate interface is discretized into a mesh with 4062 triangular elements with an average side length of 12 km. We consider several alternative loading scenarios with different spatial distributions and sizes of the asperities inferred from historical earthquake rupture asperities and recent ones. In these scenarios, asperities are simulated by applying a constant back-slip rate (relative plate convergence rate of 65 mm/yr[74]) boundary condition on the locked fault patches, allowing the surrounding patches of the fault interface to slip under a zero shear-traction condition, with no fault-normal displacement discontinuity, as a consequence of the loading imposed by the backward slip. For simplicity, we use a constant value of relative plate convergence rate and ignore its modest along-trench variation (Fig. 1). The forward model results are compared to GPS observed interseismic velocities in a North America fixed reference frame from Li & Freymueller[22], after correction for the trench-parallel motion of the Alaska Peninsula block using the re-estimated angular velocity of the block[22]. We evaluate the goodness of fit between the observed and predicted velocities using the normalized $\chi^2$ misfit statistic expressed as

$$\chi^2 = \frac{1}{2N} \sum_{i=1}^{2N} [(\text{obs}_i - \text{mod}_i)/\sigma_i]^2 \tag{5}$$

where $N$ is the number of GPS stations. The $\text{obs}_i$, $\text{mod}_i$ are the observed and predicted horizontal velocities, and $\sigma_i$ are the corresponding uncertainties, at the $i$th GPS station. We exclude GPS observations on and around Mt. Veniaminof (inset of Fig. 5), which were affected by volcano inflation[24,75], when we evaluate the normalized $\chi^2$.

## Data availability

The GPS position time series used in this study were from Nevada Geodetic Laboratory (https://geodesy.unr.edu), and the interseismic GPS velocities were available from the Supporting Information of Li & Freymueller (2018) (https://agupubs.onlinelibrary.wiley.com/doi/full/10.1002/2017GL076761). Focal mechanisms and aftershock catalog are provided by the GCMT (https://www.globalcmt.org) and USGS (https://earthquake.usgs.gov/earthquakes/), respectively. The finite coseismic slip model of the 2021 Chignik earthquake is from USGS (https://earthquake.usgs.gov/realtime/product/finite_fault/ak0219neiszm_2/us/1628016150045/complete_inversion.fsp, last accessed August 22, 2021). The GPS-derived coseismic and postseismic data and the preferred interseismic backward slip rate distribution are available in a public repository (https://doi.org/10.5281/zenodo.6443632).

## Code availability

Codes for kinematic slip inversion and for stress-driven frictional afterslip evolution may be requested from the authors.

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

## Acknowledgements

We thank Sylvain Barbot for sharing the software package Unicycle, which was rewritten in python programming language to simulate the stress-driven frictional afterslip models in this work. Most figures were prepared using the Generic Mapping Tools (GMT) software. We thank Lingling Ye, Zhuohui Xiao, Chengli Liu for providing their coseismic slip models. We also thank Kefeng He for testing the kinematic afterslip models and Jeff Freymuller for his comments on an earlier version of this manuscript. B.Z. is supported by National Natural Science Foundation of China (No. 42074116) and by Spark Program of Earthquake Science (No. XH21020). R.B. acknowledges support by NSF award EAR-1801720.

## Author contributions

B.Z. conceived this study and did most of the modeling. B.Z. and R.B. wrote the paper and interpreted the modeling result. D.W. and J.Y. conducted the analysis of the GPS data. J.Z. generated and meshed the slab interface. Q.L. helped to analyze the aftershock sequence.

## Competing interests

The authors declare no competing interests.
