## [Peer Review File · Nature Communications]

REVIEWER COMMENTS

Reviewer #1 (Remarks to the Author):

This study used GPS data to invert the coseismic slip and 87-day afterslip distribution for the 2020 Mw 7.8 Shumagin earthquake and infer the interseismic asperity coupling using the backslip model. Overall, the paper is well written and comprehensive about the 2020-2021 earthquake sequence in the Shumagin seismic gap. The first-order pattern of coseismic slip distribution is consistent with other published slip models using GPS, seismic and tsunami data, albeit with different details. Given different assumptions and modeling strategies, afterslip and asperity coupling models are dramatically different from other published models. So, it is worthwhile to publish, but some direct comparison and further modeling are required to confirm their findings.

I am very curious why the afterslip model is so different from that by Crowell and Melgar (2020, GRL), even for the 10-day solution. Since the accumulative afterslip decay with time rather rapidly, the first 10-day solution (peak afterslip of 0.4 m in C&M) should be similar to the 87-day solution in this study (peak kinematic afterslip of 0.4 m in this study) by using the same data. Whether there exists afterslip updip is critical given the uncertain potential for the shallow tsunami earthquake. Only $t=10$ days snapshot (what time interval for the snapshot?) is shown in Figure 3, I suggest authors give more direct comparisons: 1) inverting for the 10-day accumulative afterslip, same as that in C&M paper; 2) comparing the data fit from both C&M model and the model in this study to understand why different.

Afterslip inversions with the zero-slip constraint show that afterslip tends just to surround the main slip area, so it seems that the result strongly depends on the coseismic model. I wonder how the zero-slip region is chosen (slip > 2.0 m)? What is the criterion? A variety of coseismic slip models have been published, such as Crowell & Diego (2020), Liu et al. (2020), and Ye et al. (2020). They all used static GPS data but with more data constraints, such as strong-motion data, tsunami data, etc. Given the dependence of coseismic slip distribution on the afterslip model, the Authors should compare their coseismic models with those models directly and then show the inverted afterslip models with zero-slip constraint based on those models. The reason is that those coseismic slip models with more data constraints show more slip complexity than the model in this study.

Is the rake angle fixed at 90 deg for all inversions? In Figure 5, there is a systematical difference in the vector distribution between the predicted and observed interseismic GPS data. I wonder if that is due to some oblique slip?

Some minor suggestions:

1) Figure 3: Same color pattern with different ranges for stress-driven afterslip and kinematic slip is confusing (hard to understand), especially given the ambiguous caption about (F); what is the time interval of slip for (A-D)? It might be good to split into two figures, one showing snapshots from the stress-driven afterslip distribution and the other with accumulative afterslip for stress-driven afterslip, kinematic afterslip, and results without zero-slip constraint.

2) Given the GPS data distribution, the constraint of coupling/backslip rate for the 1946 zone is quite limited. I suggest the authors limit the study area just to the east of about 162W. It might add some misleading for readers given the general interest in the 1946 tsunami earthquake.

3) Supplementary Figures 3 and 4: labels for color bars might have a wrong unit, m -> mm?

Reviewer #2 (Remarks to the Author):

This is a timely and important study. It provides improved information about afterslip for the 2020 Shumagin Mw 7.8 earthquake and it provides an alternate representation of

the geodetic locking of the megathrust both preceding and subsequent to the 2020 Shumagin and 2021 Chignik earthquakes, that radically changes the impression conveyed in about 5 papers on the geodetic locking by Freymueller's group. The paper is well written and sensible in conveying the limitations, non-uniqueness and context of the models, and it is acceptable as is or with very minor revisions.

The notion of 'persistent' in the title is central to the value of the paper; typically this is evaluated by observation of repeated rupture of the same portion of a fault with overlapping slip zones, which is actually rarely possible for large events due to the long time interval between them and the limitations of historic recordings of earlier events, when they are available at all. I have found as many cases of 'repeated' large ruptures that in detail do not appear to have the same slip zone as earlier event (or at least only partial overlap) as there are cases of well-resolved repeated similar slip of a given region which convincingly indicates persistence through at least two cycles. Here, the inference is made that slip zones of past events, including the 1964, 1938, 1948 and 1946 events have concentrated zones of slip-deficit accumulation (hence persistent re-locking of prior failure), and the 2020 and 2021 slip patches, which do not overlap those ruptures, had produced prior localized slip deficit as well. 2020 may have ruptured in 1917, in what appears to be a smaller event, but details of the slip of the earlier event are very limited, so it is ambiguous how persistent that patch is. The 2021 event occurred within the poorly defined 1938 aftershock region, but does not have significant overlap with the large slip patch(es) in the earlier events. So, here the case is not made from observed repeated rupture of a given persistent patch, but overlap of slip zones and plausible subsequent slip deficit zones. This leads to the summary figures 5 and 7, which show very localized asperities, with only 1917/2020 possibly being a re-ruptured persistent asperity. The assumption that one can match the slip deficit constraints (which are mostly from distant GNSS stations other than below the Shumagin Islands and below the Semidi Islands) by assuming concentrated slip deficit patches is demonstrated to be viable, but it is not unique, given that prior studies in multiple papers have matched the same geodetic data as well with laterally segmented, depth varying locking models with maximum slip deficit near the toe of the wedge. I personally much prefer the asperity based models, as they are a more parsimonious representation of the locking pattern and they exploit the information from actual coseismic large-slip patches, and I am comfortable with likely persistence of those patches, but the uniqueness remains troubling. As a non-geodesist interested in locking models, I cannot help but be frustrated by having such different and fundamentally discrepant representations of present geodetic locking. But this makes it all the more important to balance the misleading 'consistency' of multiple prior, redundant studies that assumed a particular parameterization of locking.

Given the clear lack of resolution of the shallow portion of the megathrust, I did not attach much significance to the scattering of small unresolved patches shown in Figure 7, and the authors might want to rethink the added detail relative to Figure 5 (yes, it is plausible, but such tiny patches could be everywhere, including at shallow depths; perhaps better to keep the summary figure close to the data. I would argue that the tsunami modeling for 1938 provides, at best, a very weak basis for the western patch in 1938 as well, and it is the case that Freymueller et al. (2021) provide a range of slip models for 1938 regional tsunami that is quite different from than Johnson and Satake (and they point out real weaknesses in the earlier tsunami studies). One might see value in a sensitivity study; remove the western small patch in 1938; can you not fit the GNSS trends? Seems you must be able to map anything there into the 2021 and 2020 zones. Similarly, 1948 is poorly located and with the strong down-dip edge of slip in 2020 and 2021 being above 40 km depth, I am skeptical that rupture in 1948 is as deep as needed to lie on SLAB2 (rupturing from 40-50 km). If it were confined to above 40 km depth, I expect the GNSS data can be fit fine, as it is a small localized patch in a region of low coupling. Essentially, I am encouraging a bit more exploration of how much this is "you get out, what you put in". I am OK, with 1946 being near the trench and pretty much unconstrained; if you take away that patch does it really matter? My models to make the far-field tsunami large enough for 1946 with a 'tsunami-earthquake' are requiring near-trench slip of 50-100 m, so it may not have significant slip-deficit accumulation relative to prior failure. Anyhow, if you are going for a 'parsimonious' representation of coupling, the approach could be to do a bunch of 'leave one out/move to plausible location' tests for the 7 patches to see which are really controlled by data for the parsimonious

representation versus just a result of allowing them to exist. The paper included consideration of additional patches, finding that sure they could exist too (notably for the updip region in Shumagin gap region), and I am sure there could be a lot more (given that laterally extensive locking zones can work), but the question is what is the least complexity you actually need to achieve a reasonable fit. My guess is that you could toss out 1946 and the western patch of 1938 and not degrade the fits.

Very minor things::;

line 35 "normal" is a confusing term because it confounds with normal-faulting. Perhaps "typical" or "conventional" is less ambiguous.

line 50-52. Well, there are seismic gaps with no known historical failure and there are seismic gaps known to have had prior rupture. No question about that. Many seismologists distinguished these cases by using "Seismic Gap" explicitly for regions with known large prior events for the purpose of recurrence estimation and coupling assessment, noting ambiguous regions as "possibly aseismic gap", etc.. It is not clear that many of these region are "weakly coupled" or not; geodetic control is lacking on slip-deficit for major 'seismic gap' regions like Izu, Marianas, central Tonga, Southern Kermadec, etc., or is remote for places like Tehuantepec. I would at least remove the word "still" from line 52, as it is confusing (we don't know what we don't know). I believe that given a few more years and with broader recognition of the on-land geodetic slip-deficit estimates and incorporation of the seafloor geodesy constraint, the Tohoku region would have been redefined as not being weakly coupled (indeed, the historical seismic record had suggested that seismic coupling might be low). So, while I agree that regions truly known to be weakly coupled are interesting, regions for which we have no real constraint are equally important targets for "better understanding the slip behaviors during earthquake cycles". The 1938 zone is not 'weakly coupled', but your own study shows how valuable it is to know the slip behavior for at least two large events in the area.

line 108. The 2005 event was Mw 8.7; that is rather a poor comparison to make with the 7.8 in the 2020. Hard for such a small event to have ruptured to the trench. Yes, all models suggest that it did not, but the better comparison would be with a 7.8 that ruptured depths of 20-40 km and also reached to the trench, and I cannot think of a case. So, the comparison is not very informative.

line 219. Given my skepticism about resolution, I have a hard time believing that one can resolve the locking area to be only 25% of the coseismic slip area. Yes, the preferred models do that, but is it not a consequence of some effort in the inversion to find a minimum area intrinsically? I have no problem with the physical plausibility of this finding, but am dubious that one has real resolution of pin-point locking. Not sure what to recommend other than perhaps some discussion so that readers know what to believe versus what is an inversion bias. My unease comes from comparing the 'kinematic' and 'stress-driven' afterslip distributions seen in Figure 3; the highly parameterized 'physical' model gives results that are very different in absolute slip placement from the kinematic solution. This shows that the data can be fit by such different models that it is frustrating (that is just the way it is), and it makes me doubt any claims of precise resolution of locking areas.

line 340-341 the sentence here is confusing: and needs some clarification: "After subtracting the moment released during the 2021 Chignik earthquake in this zone, which is approximately 50% of the total moment of the 2021 event, the remaining cumulative moment deficit is estimated to be 2.9×10^{21} Nm (Mw=8.2)."

Overall, this is excellent work, and I encourage consideration of some of my issues, but only minor revisions are needed for this to be an acceptable contribution in Nature Communications. Thorne Lay.

Reviewer #3 (Remarks to the Author):

In their paper "Aseismic slip and recent ruptures of persistent asperities along the Alaska Aleutian subduction zone" Zhao et al. present a co-seismic and early post-seismic slip model (based on kinematic and stress driven modeling) for the 2020 M7.8 Simeonof earthquake. They link their findings to slip from the recent 2021 M8.2 Chignik earthquake and other large events in the Alaska-Aleutian subduction zone to infer a conceptual model of heterogeneous coupling distribution, and stable asperities in this region. They argue that the latter are stable.

This is an interesting paper that brings a different dimension to the existing, purely kinematically derived interseismic coupling models for the region. The findings for the afterslip distribution, both from kinematic and stress driven modelling are an important update to the findings that exist so far (Crowell and Melgar, 2020), which were based on very limited data. The data analysis and modeling methodologies are sound, the interpretations generally robust. While I have a few questions and concerns that I outline below, I do think the timeliness and broad impact of the paper warrant publication in Nature Communications after minor revisions.

My main concern with the paper is the suggestion that these asperities are persistent. There's ample evidence for significant variability in terms of rupture size in the general region (e.g., Shennan et al., 2009 & 2014; Briggs et al., 2014) and I don't quite understand how this meshes with the proposed persistent asperity hypothesis. The main difference between the existing literature (e.g., Li & Freymueller; Drooff & Freymueller) is that the current paper proposes a rather heterogeneous along dip segmentation of coupling, based on historic and recent events (and some trial and error modeling). They recognize that the along dip coupling is difficult to impossible to constrain with the existing data. So, instead of having to justify a regularization strategy that partitions the along dip coupling, they lean on locations of events. The trouble with those is that they need to have happened first and unfortunately, the authors cannot either provide much certainty on the characteristics of the shallow interface. Remarkably, their results generally match the along-strike segmentation, which remains quite stable in different approaches of kinematic coupling inversions using the GNSS data. This, I find is the important finding of their analysis. In light of this, I would appreciate if the authors would more prominently discuss the assumptions of their asperity model and what their reasons are to assume long-term persistence of these.

The other major comment I have is regarding the stress-driven afterslip model. Yes, the authors are correct that this is more physical, but the fit to the data, particularly in the near field and down-dip their rupture zone (AC28, AV07, AC25, AC21) is not good. This can have several reasons. One - this model is driven by the co-seismic slip distribution which is derived solely from GNSS data, but those are relatively sparse in the area. So this postseismic misfit could represent issues with the co-seismic model. Two - assumptions made for the parameters governing the postseismic slip are not good. The fact that the model overpredicts uplift at AC28 and subsidence at AV07 signifies that the amount of afterslip in the downdip region is too large (it is not clear to me whether the afterslip is purely thrust, too, or has a strike slip component to it). Since the authors lean heavily on this model throughout the paper, I would request they either optimize the fit, or provide a more detailed discussion on why these misfits are acceptable. The current mention in L160-161 that "the position time series from this afterslip model are also in good agreement with the observations (Fig. 4)." is a bit too optimistic. Furthermore, it would be instructive to explain why the stress driven model suggests such localized, high amplitude afterslip, whereas the kinematic data-driven inversion requires much more distributed, but lower amplitude slip.

Not as important as these first two points, but worth mentioning before the minor comments is the statement in L289 where a relationship between afterslip and aftershocks is drawn. "This suggests that the aftershocks in the first 87 days may be mainly triggered by aseismic afterslip in the peripheral region of the coseismic area." This seems to jump the gun and mistake (temporal) correlation for causation. Wouldn't you want a better spatial correlation between afterslip maxima and aftershock locations? Figure 6A suggests very poor spatial correlation. If this were true, I would expect most of the aftershocks in the high afterslip regions, but we don't see this. Based on figure 3F one could make the argument that most of the aftershocks occur in a gap of downdip and updip afterslip to the west of the rupture zone, releasing the co-seismically

generated stresses seismically, rather than aseismic. The same may be roughly true for the updip cap, although that one cluster of aftershocks locates a bit too far west for that.

I consider these comments minor, as I don't expect that the major points of this paper will be significantly altered, nor should it take too long to address these points. I hope to see this published in Nature Communications in the near future!

Minor comments:

L32 "earthquake" delete "s"

L35 "normal earthquakes" seems ambiguous in the context "regular"?

L248: Invoking a slow slip event here seems maybe a step too far. What is the evidence for this? And how does afterslip differ from a slow slip event in this case?

L267 "We suspect that the poor correlation between the spatial distribution of the aftershocks and afterslip is in part caused by large uncertainties in the aftershock locations" This seems at odds with the comment about a good spatial correlation between the two just lines before this. Could you at least give an estimate of the location error?

L293: "The seismic moment released by the aftershocks in the 87-day postseismic period is about two orders of magnitude smaller than the moment released by aseismic afterslip (Supplementary Fig. 14)." What does this tell us? This statement isn't picked up anywhere.

L327 Is the fit really equally good?

L331 makes the same argument that the existing kinematic coupling models make - there's no data near the trench, so we don't know. The difference is that the coupling in the Li & Freymueller, and Drooff & Freymueller models is uniformly larger, rather than asperity driven, which we only know about after the fact.

L339 - What is the "red curve" in figure 5?

L422 - Why no strike slip component in the slip model? Other models have shown moderate (Liu et al, 2021) to significant (Ye et al, 2021) along strike motion.

L465 - specifically, what are the burn in and thinning parameters applied? What's the autocorrelation of the remaining samples?

References:

I. Shennan, R. Bruhn, G. Plafker, Multi-segment earthquakes and tsunami potential of the Aleutian megathrust, *Quaternary Science Reviews*, 28(1-2), 7-13 (2009).

Shennan, I., Barlow, N.L.M., Carver, G.A., Davies, F.P., Garrett, E., Hocking, E.P. (2014). Great tsunamigenic earthquakes during the last 1000 years on the Alaska megathrust, *Geology*, 42(8), 687-690

R. W. Briggs, S. E. Engelhart, A. R. Nelson, T. Dura, A. C. Kemp, P. J. Haeussler, D. R. Corbett, S. J. Angster, L. A. Bradley, Uplift and subsidence reveal a nonpersistent megathrust rupture boundary (Sitkinak Island, Alaska), *Geophysical Research Letters*, 41(7), 2289-2296 (2014).

Reviewer #1 (Remarks to the Author):

This study used GPS data to invert the coseismic slip and 87-day afterslip distribution for the
2020 Mw 7.8 Shumagin earthquake and infer the interseismic asperity coupling using the
backslip model. Overall, the paper is well written and comprehensive about the 2020-2021
earthquake sequence in the Shumagin seismic gap. The first-order pattern of coseismic slip
distribution is consistent with other published slip models using GPS, seismic and tsunami
data, albeit with different details. Given different assumptions and modeling strategies,
afterslip and asperity coupling models are dramatically different from other published
models. So, it is worthwhile to publish, but some direct comparison and further modeling are
required to confirm their findings.

Thanks for your positive evaluation and suggestions. As described below, we expanded on
the comparison with other models.

I am very curious why the afterslip model is so different from that by Crowell and Melgar
(2020, GRL), even for the 10-day solution. Since the accumulative afterslip decay with time
rather rapidly, the first 10-day solution (peak afterslip of 0.4 m in C&M) should be similar to
the 87-day solution in this study (peak kinematic afterslip of 0.4 m in this study) by using the
same data. Whether there exists afterslip updip is critical given the uncertain potential for the
shallow tsunami earthquake. Only $t=10$ days snapshot (what time interval for the snapshot?)
is shown in Figure 3, I suggest authors give more direct comparisons: 1) inverting for the 10-
22 day accumulative afterslip, same as that in C&M paper; 2) comparing the data fit from both
C&M model and the model in this study to understand why different.

Thanks for your critical suggestions, which we address below and allowed us to provide more
details on the afterslip modeling in the manuscript.

There are several differences in the methods of afterslip inversion between our own models
and that of Crowell and Melgar (2020). (1) We only inverted for the dip-slip component
(rake=90 deg), while the rake in C&M (2020) was constrained to be bounded between 45
and 135 deg. (2) We employed a scale-dependent umbrella operator to approximate the
discrete Laplacian (Maerten et al., 2005, Maerten05), while C&M (2020) used the zeroth-
order Tikhonov (minimum norm) approach. (3) Another difference is the method of
computing the Green's functions. We calculated the Green's functions using triangular

dislocation patches (Nikkhoo and Walter, 2015), while C&M (2020) utilized the frequency-
 wavenumber integration method integrated in the MudPy code they used.

 Figure R1. Kinematic afterslip models with different rake constraints and regularization
 methods and time intervals.

Therefore, we tested a series of kinematic afterslip models with different rake constraints and
 regularization methods as well as different time intervals of the inverted dataset. We tested
 different afterslip models with the same triangular mesh that was provided by Crowell but

without zero-slip constraint. All these models well fit the observed GPS observations (Fig.
R1). When the rake is fixed at 90 deg (only dip slip) and the Maerten05 regularization is
used, the inverted afterslip mainly appears on the updip and western portions of the coseismic
rupture zone with minor afterslip on the downdip portion (Model 1, Fig. R1a). The spatial
distribution of afterslip is similar to the result in C&M (2020) when the Tikhonov
regularization method is used, although the peak afterslip is shallower than in their model
(Model 2, Fig. R1b). When the rake is constrained to vary between [45, 135] deg as assumed
in C&M (2020), the peak afterslip with regularization method Maerten05 (Model 3, Fig. R1c)
mainly occurs at shallow depths. The afterslip model with the regularization method of
Tikhonov (Model 4, Fig. R1d) shares a similar distribution with that of Model 2 (Fig. R1b),
albeit there is some overlap with the coseismic rupture zone. These test results show that the
rake constraint and regularization method indeed influence the inverted distribution of
afterslip, especially when the number of observations is limited and the displacement
magnitude is small. The test results also indicate that the afterslip inversions are robust when
using the Tikhonov regularization method, no matter how the rake is constrained. However,
these aforementioned factors do not affect the coseismic slip models too much (see below).
We also examined whether the time interval of observations impacts the afterslip distribution.
We obtain similar afterslip distributions when using 10-day observations with the same rake
constraint and regularization method as C&M (2020) used (Model 5, Fig. R1e), suggesting
that the afterslip on the slab interface evolves with time. Although the obtained afterslip
models with the Tikhonov regularization method become similar to the results in C&M
(2020), some differences still exist in detail. For example, we can observe obvious updip
afterslip beneath station AC12. In addition, we notice that the peak afterslip in C&M (2020)
is nearly twice that of our own model in Fig. R1e. This discrepancy is probably due to the
differences in other parameters, such as the Green's functions, data weighting and smoothing
factor.

Following the above tests, we reinvert for kinematic afterslip models with zero-slip constraint
using our own mesh with the Tikhonov regularization method (see below) and replot Figure 3
in the manuscript. We have documented explicitly that we used the Tikhonov regularization
method in the kinematic afterslip inversion in Lines 615-618 (tracked PDF file), as shown
below:

“However, here we employed the zeroth-order Tikhonov regularization method, since we find
it is more robust than the one used in the coseismic slip inversion when the number of
observations is limited and the displacement magnitude is small (Supplementary Fig. 5)”

We have rewritten the comparison of different afterslip models in Lines 324-328 (tracked
PDF file). The sentences read “While our coseismic slip distribution is generally in
agreement with that of Crowell & Melgar²⁷, differences exist between our kinematic and
stress-driven afterslip models and theirs. The kinematic afterslip model of Crowell &
Melgar²⁷ features a majority of afterslip downdip of the coseismic rupture, although they
indicate that the existence of updip afterslip cannot be ruled out. In contrast, our inferred
afterslip model from the kinematic inversion with zero-slip constraint suggests afterslip
surrounding the coseismic peak slip zone (Fig. 3a).”. Further comparison of different
afterslip models can be found in Lines 368-372 (tracked PDF file), as shown below:

“Kinematic afterslip inversions without zero-slip constraint and using the same triangular
mesh and rake constraint as used in Crowell & Melgar²⁷ also show significant updip
afterslip, as well as downdip afterslip in the west of the coseismic rupture area
(Supplementary Fig. 5). We suspect the differences can attributed to the method of
calculating the Green’s Functions. Different from the triangular dislocation method we used,
Crowell & Melgar²⁷ employed a point-source representation.”

We added Fig. R1 as Supplementary Fig. 5 and documented the afterslip models without the
zero-slip constraint in Lines 127-129 (tracked PDF file). It now reads “If the zero-slip
constraint in the coseismic zone is not applied, the afterslip also appears on the periphery of
the coseismic rupture with the majority of afterslip in the western part when using the zeroth-
order Tikhonov regularization approach (Supplementary Fig. 5)”.

1. Afterslip inversions with the zero-slip constraint show that afterslip tends just to surround
the main slip area, so it seems that the result strongly depends on the coseismic model. I
wonder how the zero-slip region is chosen (slip > 2.0 m)? What is the criterion? A variety of
coseismic slip models have been published, such as Crowell & Diego (2020), Liu et al.
(2020), and Ye et al. (2020). They all used static GPS data but with more data constraints,
such as strong-motion data, tsunami data, etc. Given the dependence of coseismic slip
distribution on the afterslip model, the Authors should compare their coseismic models with
those models directly and then show the inverted afterslip models with zero-slip constraint

based on those models. The reason is that those coseismic slip models with more data
constraints show more slip complexity than the model in this study.

Thanks for your suggestion.

We apologize that we did not describe in detail how we determine the zero-slip region in the
main text. Following previous works, we tested and chose the critical slip value by trial and
error. We have revised the manuscript to clarify this procedure.

Figure R2. Comparison of different coseismic slip distributions.

Yes, several coseismic slip models constrained with more data constraints have been
published by now, including the most recent paper by Xiao et al. (2021, EPSL). We compare
our model to those published models in Fig. R2 (now Supplementary Fig. 20). To first order,
our coseismic rupture zone overlaps with the others, but some differences are also obvious.

The peak slip area of our model is located further east compared to other models. In addition,
our model does not have distinct small asperities. However, we find that the kinematic
afterslip models with zero-slip constraint and stress-driven afterslip based on these complex

coseismic models produce similar fit to GPS observations compared to models with our
preferred coseismic model. These test results are shown below (Fig. R3).

Figure R3. Kinematic afterslip models with the zero-slip constraint based on different
coseismic slip models with different rake constraints. The WRMS misfit for each model is
labelled in each panel.

Figure R3. Continued.

We have conducted a series of kinematic afterslip inversions with the zero-slip constraint
 using those different coseismic slip models following your suggestion. We assessed different
 critical slip values by trial and error for different coseismic models, and we also considered
 the effect of the rake constraints in these models. Fig. R3 shows the afterslip distributions
 with different rake constraints based on different coseismic slip models. The distribution of
 the inverted afterslip varies with the used coseismic models, but is mostly insensitive to the
 chosen rake constraint (Fig. R3). The cumulative 87-day postseismic displacements can
 be equally well fit by all models. The WRMS value between the observed and modeled
 postseismic displacements varies from 1.9 to 2.2 mm. In all the models, the afterslip occurs
 around the periphery of the coseismic rupture but never on the downdip extent alone.

To document this analysis, we added a sentence in the *Methods* section in Lines 624-625
 (tracked PDF file), “*We test the performance of kinematic afterslip models with zero-slip*
 *constraint based on other published coseismic models and models allowing for a strike-slip*
 *component*”. We documented the finding in Lines 124-127 (tracked PDF file) as shown
 below:

“We find that the afterslip models with zero-slip constraint based on the various coseismic
models^{27–30} produce similar data fits to the GPS observations, and are mostly insensitive to
the chosen rake constraint (Supplementary Fig. 4).”

We have added Fig. R2 (now Supplementary Fig. 20) and Fig. R3 (now Supplementary Fig.
4) in the supplementary materials. We reinverted for the afterslip distribution with zero-slip
constraint based our own original coseismic model and updated Figure 3 in the manuscript.
We added a sentence in Lines 615-618 (tracked PDF file), “However, here we employed the
zeroth-order Tikhonov regularization method, since we find it is more robust than the one
used in the coseismic slip inversion when the number of observations is limited and the
displacement magnitude is small (Supplementary Fig. 5)”.

Is the rake angle fixed at 90 deg for all inversions? In Figure 5, there is a systematical
difference in the vector distribution between the predicted and observed interseismic GPS
data. I wonder if that is due to some oblique slip?

Figure R4. Comparison of coseismic slip models with different rake constraint and
regularization methods. The regularization method of Maerten05 is used in **a** and **c**, and the
Tikhonov approach is used in **(b)** and **(d)**. The WRMS misfit for each model is labelled in
each panel.

Yes, we only consider pure thrust slip along the Alaska-Aleutian subduction zone with rake
angle fixed at 90 deg in all models, including the coseismic, afterslip and interseismic
asperity models. This assumption is reasonable since the focal mechanism solutions of
earthquakes around the slab interface indicate a predominance of thrusting faulting (e.g., Fig.
1 in Ye et al., 2021). To examine the performance of models considering the strike-slip
component, we allowed the rake to vary between 45 and 135 deg rather than fixed at 90 deg
when inverting for both coseismic and afterslip models. The obtained coseismic slip
distribution is to first-order similar to previous results with more data constraints, although
the peak slip location moves somewhat (Fig. R4). It is also noticeable that the maximum
coseismic slip in models with the Tikhonov regularization method is smaller than those with
the other regularization method.

To document the analysis of the coseismic models with a strike-slip component, we now state
the test method in Lines 609-611 (tracked PDF file), "*We also test the performance of*
*coseismic models with a strike-slip component (rake allowed to vary between 45°~135°) and*
*with another regularization method, the zeroth-order Tikhonov (minimum norm) approach.*".
The findings can be found in Lines 109-118 (tracked PDF file), "*Tests allowing for a strike-*
*slip component in the coseismic models and with a different regularization method show*
*similar slip distributions, although the peak slip magnitude and location changes somewhat*
*(Supplementary Fig. 3).*". Fig. R4 (now Supplementary Fig. 3) is added in the supplementary
materials.

No matter what rake constraint is applied, the inverted afterslip models share similar
distribution patterns and the dip slip is the primary component (Fig. R3). The findings from
testing of kinematic afterslip models in Lines 124-127 (tracked PDF file), "*We find that the*
*afterslip models with zero-slip constraint based on the various coseismic models²⁷⁻³⁰ produce*
*similar data fits to the GPS observations, and are mostly insensitive to the chosen rake*
*constraint (Supplementary Fig. 4)*".

We also tested whether the fit between the observed postseismic time series and predictions
from stress-driven afterslip models can be improved, when using the other coseismic slip
models (Fig. R5 and R6) and our own test model shown in Fig. R4c with rake constraint
between [45, 135] deg. (Fig. R7). We extended the length and width of these coseismic fault
planes to model the afterslip evolution driven by stress. We here only present the results
assuming that the frictional parameter and reference velocity are uniform over the whole
afterslip zone, because the undip and downdip afterslip models all yield relatively large
normalized χ^2 . We eventually find that our coseismic model in the main text with fixed rake
at 90 deg can fit the coseismic offsets as well as the postseismic deformation with the
smallest misfit value.

We documented the test method of the stress-driven afterslip models by inserting sentences in
Lines 679-683 (tracked PDF file), “We also test whether the stress-driven frictional afterslip
models based on other coseismic models^{28,30} and our own test model with a coseismic strike-
slip component can improve the fit between the observed and modeled postseismic time series
applying the same procedure as described above. We extended the length and width of the
modeled fault planes in Liu et al.²⁸ and Xiao et al.³⁰ to cover a wider region on which
afterslip is allowed to occur”. We added the findings in Lines 207-210 (tracked PDF file),
“Frictional afterslip models, which are based on other coseismic models^{28,30}, all yield
relatively large normalized χ^2 values (4.2~4.5, Supplementary Figs. 9-11) compared to our
preferred model (normalized $\chi^2=3.0$). Models invoking a coseismic strike-slip component
also cannot improve the data fit (Supplementary Figs. 9-11)”.

As for the interseismic asperity model, the systematic differences in azimuth at GPS sites
above the 1938 and 1964 rupture zone between the observed and modeled interseismic GPS
velocities probably reflect the complex forearc deformation (Elliott & Freymueller, 2020;
Freymueller, personal communication, 2021) rather than oblique slip on the slab interface.
Focal mechanisms of earthquakes along this segment also show predominantly pure thrust
events. Therefore, we did not consider the strike-slip component in the interseismic asperity
model. We now add the citation of Elliott & Freymueller, 2020 as ref. 36 in Line 272 of the
tracked PDF file.

Figure R5. Stress driven afterslip model based on Liu et al. (2020)'s coseismic slip model. **a** and **b** show the cumulative 87-day afterslip distributions with rake fixed at 90 deg and with rake the same as that in the coseismic model, respectively. **c** Illustrates the comparison between GPS observed and modeled postseismic time series. Models in **a** and **b** yield normalized χ^2 of 4.6 and 4.2, respectively, both are larger than that of the preferred model (normalized $\chi^2=3.0$) in the manuscript.

Figure R6. The same as Fig. R5 but with Xiao et al. (2021)'s coseismic slip model. Models in
 **a** and **b** yield normalized χ^2 of 4.6 and 5.5, respectively.

Figure R7. The same as Fig. R5 but with the coseismic slip model shown in Fig. R4c. Models
 in a and b yield normalized χ^2 of 4.9 and 4.3, respectively.

Some minor suggestions:

1) Figure 3: Same color pattern with different ranges for stress-driven afterslip and kinematic
 slip is confusing (hard to understand), especially given the ambiguous caption about (F); what
 is the time interval of slip for (A-D)? It might be good to split into two figures, one showing
 snapshots from the stress-driven afterslip distribution and the other with accumulative
 afterslip for stress-driven afterslip, kinematic afterslip, and results without zero-slip
 constraint.

We used different color scales for stress-driven afterslip and kinematic afterslip because their
maximum values differ substantially. If we used the same color scale, it would be hard to see
the kinematic afterslip distribution. We now added the words “Note the different color scales
in the stress-driven vs. kinematic afterslip models.” in the caption of Fig. 3 to make sure
readers don’t misinterpret the results. We decided to only show the cumulative kinematic and
stress-driven afterslip distributions in Fig. 3, given that showing the time-dependent afterslip
evolution is not very informative. We have updated this figure.

2) Given the GPS data distribution, the constraint of coupling/backslip rate for the 1946 zone
is quite limited. I suggest the authors limit the study area just to the east of about 162W. It
might add some misleading for readers given the general interest in the 1946 tsunami
earthquake.

We still favor a relatively large coverage to show the heterogeneous along-strike distribution
of asperities. As shown in Supplementary Fig. 8 (now Supplementary Fig. 12), an end-
member test indicates that the GPS stations located on Unimak Island would record high
interseismic velocities if a relatively large asperity for the 1946 zone is invoked. This
supports the idea that the low magnitude of interseismic velocities on Unimak Island reflects
small-sized asperities along this segment. A test without any locked asperity in this area
(Supplementary Fig. 13, now Supplementary Fig. 17) still fits the data quite well. We
emphasize the limited resolution of our model and note in Line 285 (tracked PDF file) that “*If*
*this asperity is excluded, the misfit between the observed and predicted velocities decreases*
*slightly (Supplementary Fig. 17), and it is possible that the actual rupture asperity of this*
*event lies further west in an area lacking GPS coverage.*”

3) Supplementary Figures 3 and 4: labels for color bars might have a wrong unit, m -> mm?
Thanks for your careful reading. We have substituted these two figures with Figs. R4 and R3,
respectively.

Reviewer #2 (Remarks to the Author):

This is a timely and important study. It provides improved information about afterslip for the
2020 Shumagin Mw 7.8 earthquake and it provides an alternate representation of the geodetic
locking of the megathrust both preceding and subsequent to the 2020 Shumagin and 2021

Chignik earthquakes, that radically changes the impression conveyed in about 5 papers on the
geodetic locking by Freymueller's group. The paper is well written and sensible in conveying
the limitations, non-uniqueness and context of the models, and it is acceptable as is or with
very minor revisions.

Thanks for your positive evaluation.

The notion of 'persistent' in the title is central to the value of the paper; typically this is
evaluated by observation of repeated rupture of the same portion of a fault with overlapping
slip zones, which is actually rarely possible for large events due to the long time interval
between them and the limitations of historic recordings of earlier events, when they are
available at all. I have found as many cases of 'repeated' large ruptures that in detail do not
appear to have the same slip zone as earlier event (or at least only partial overlap) as there are
cases of well-resolved repeated similar slip of a given region which convincingly indicates
persistence through at least two cycles. Here, the inference is made that slip zones of past
events, including the 1964, 1938, 1948 and 1946 events have concentrated zones of slip-
deficit accumulation (hence persistent re-locking of prior failure), and the 2020 and 2021 slip
patches, which do not overlap those ruptures, had produced prior localized slip deficit as
well. 2020 may have ruptured in 1917, in what appears to be a smaller event, but details of
the slip of the earlier event are very limited, so it is ambiguous how persistent that patch is.
The 2021 event occurred within the poorly defined 1938 aftershock region, but does not have
significant overlap with the large slip patch(es) in the earlier events. So, here the case is not
made from observed repeated rupture of a given persistent patch, but overlap of slip zones
and plausible subsequent slip deficit zones. This leads to the summary figures 5 and 7, which
show very localized asperities, with only 1917/2020 possibly being a re-ruptured persistent
asperity. The assumption that one can match the slip deficit constraints (which are mostly
from distant GNSS stations other than below the Shumagin Islands and below the Semidi
Islands) by assuming concentrated slip deficit patches is demonstrated to be viable, but it is
not unique, given that prior studies in multiple papers have matched the same geodetic data as
well with laterally segmented, depth varying locking models with maximum slip deficit near
the toe of the wedge. I personally much prefer the asperity based models, as they are a more
parsimonious representation of the locking pattern and they exploit the information from
actual coseismic large-slip patches, and I am comfortable with likely persistence of those
patches, but the uniqueness remains troubling. As a non-geodesist interested in locking

models, I cannot help but be frustrated by having such different and fundamentally discrepant
representations of present geodetic locking. But this makes it all the more important to
balance the misleading 'consistency' of multiple prior, redundant studies that assumed a
particular parameterization of locking.

We appreciate this thoughtful discussion and fully agree with (a) the characterization of
“persistence” being based on the inference of “overlap of [earthquake] slip zones and
plausible subsequent slip deficit zones” and (b) the characterization of coupling models being
“consistent but not unique” representations of the slip-deficit distribution.

We now added the following sentences in Lines 697-710 (tracked PDF file) in the *Methods*
section, “We employ an “asperity model” to infer a physically plausible distribution of
persistent asperities. It is worth noting that the considered persistent asperities are based on
the inference of overlap of earthquake rupture zones and interseismic slip deficit zones. In
this simple model, the fully coupled patches are assumed to be confined to velocity-
weakening asperities, which are fully locked during the interseismic period, and postseismic
and interseismic creep occurs on velocity strengthening areas of the fault outside of the
asperities^{9,37}.”

We emphasize the “consistent but not unique” characterization in Lines 293-308 (tracked
PDF file). It now reads “These tests suggest that the obtained characterizations of coupling
models in Fig. 5 and Supplementary Fig. 18 are consistent but not unique representations of
the slip-deficit distribution.”

Given the clear lack of resolution of the shallow portion of the megathrust, I did not attach
much significance to the scattering of small unresolved patches shown in Figure 7, and the
authors might want to rethink the added detail relative to Figure 5 (yes, it is plausible, but
such tiny patches could be everywhere, including at shallow depths; perhaps better to keep
the summary figure close to the data. I would argue that the tsunami modeling for 1938
provides, at best, a very weak basis for the western patch in 1938 as well, and it is the case
that Freymueller et al. (2021) provide a range of slip models for 1938 regional tsunami that is
quite different from than Johnson and Satake (and they point out real weaknesses in the
earlier tsunami studies). One might see value in a sensitivity study; remove the western small
patch in 1938; can you not fit the GNSS trends? Seems you must be able to map anything
there into the 2021 and 2020 zones. Similarly, 1948 is poorly located and with the strong

down-dip edge of slip in 2020 and 2021 being above 40 km depth, I am skeptical that rupture
in 1948 is as deep as needed to lie on SLAB2 (rupturing from 40-50 km). If it were confined
to above 40 km depth, I expect the GNSS data can be fit fine, as it is a small localized patch in
a region of low coupling. Essentially, I am encouraging a bit more exploration of how much
this is "you get out, what you put in". I am OK, with 1946 being near the trench and pretty
much unconstrained; if you take away that patch does it really matter? My models to make
the far-field tsunami large enough for 1946 with a 'tsunami-earthquake' are requiring near-
trench slip of 50-100 m, so it may not have significant slip-deficit accumulation relative to
prior failure. Anyhow, if you are going for a 'parsimonious' representation of coupling, the
approach could be to do a bunch of 'leave one out/move to plausible location' tests for the 7
patches to see which are really controlled by data for the parsimonious representation versus
just a result of allowing them to exist. The paper included consideration of additional patches,
finding that sure they could exist too (notably for the updip region in Shumagin gap region),
and I am sure there could be a lot more (given that laterally extensive locking zones can
work), but the question is what is the least complexity you actually need to achieve a
reasonable fit. My guess is that you could toss out 1946 and the western patch of 1938 and
not degrade the fits.

We fully agree with this assessment and it is a good idea to further explore additional
coupling scenarios, in particular related to coupling near the 1938 rupture zone. We have
already tested the scenario of no locked asperity for the 1946 event shown in Supplementary
Fig. 13 (now Supplementary Fig. 17). The test result indeed indicates that a locked asperity
associated with the 1946 rupture is not required and could be smaller or located further to the
west. We now add the proposed test of a model without the locked western 1938 asperity and
find that the model without the western asperity of 1938 indeed does not degrade the data fit
(Fig. R8). Further excluding the locked 1946 asperity does not change the data fit too much
(Fig. R9, Supplementary Fig. 18). We now document these findings in Lines 289-308
(tracked PDF file) as shown below:

“Supplementary Fig. 18 shows a plausible backslip rate (coupling) distribution with the least
complexity and only five core asperities. This simple model can also achieve a reasonable fit
since contributions due to the small and shallow asperities corresponding to the 1946
Unimak tsunami earthquake and to the western rupture of the 1938 earthquake are very
small compared to other large asperities (Supplementary Fig. 19). These tests suggest that

*the obtained characterizations of coupling in Fig. 5 and Supplementary Figs. 15-18 are*
*consistent but not unique representations of the slip-deficit distribution.”*

Figure R8. Same as Fig. 5 but without a locked western 1938 asperity. This model yields a
normalized χ^2 of 13 and a WRMS value of 2.0 mm/yr.

Figure R9. Same as Fig. 5 but without the locked western 1938 asperity and the 1946
asperity. This model yields a normalized χ^2 of 13 and a WRMS value of 2.0 mm/yr.

Figure R10. Comparison of GPS observed interseismic velocities to predictions from
individual locked asperities calculated from the forward BEM model.

Yes, we have run forward models to show surface displacement contributions associated with
locking of the individual core asperities (Fig. R10, now Supplementary Fig. 19). We have
added a sentence in Lines 290-293 (tracked PDF file), as shown below:

“This simple model can also achieve a reasonable fit since contributions due to the small and
shallow asperities corresponding to the 1946 Unimak tsunami earthquake and to the western
rupture of the 1938 earthquake are very small compared to other large asperities
(Supplementary Fig. 19)”

Very minor things:::

line 35 "normal" is a confusing term because it confounds with normal-faulting. Perhaps
"typical" or "conventional" is less ambiguous.

We have revised to “regular” earthquakes.

line 50-52. Well, there are seismic gaps with no known historical failure and there are seismic
gaps known to have had prior rupture. No question about that. Many seismologists
distinguished these cases by using "Seismic Gap" explicitly for regions with known large
prior events for the purpose of recurrence estimation and coupling assessment, noting
ambiguous regions as "possibly aseismic gap", etc.. It is not clear that many of these region
are "weakly coupled" or not; geodetic control is lacking on slip-deficit for major 'seismic gap'
regions like Izu, Marianas, central Tonga, Southern Kermadec, etc., or is remote for places
like Tehuantepec. I would at least remove the word "still" from line 52, as it is confusing (we
don't know what we don't know). I believe that given a few more years and with broader
recognition of the on-land geodetic slip-deficit estimates and incorporation of the seafloor
geodesy constraint, the Tohoku region would have been redefined as not being weakly
coupled (indeed, the historical seismic record had suggested that seismic coupling might be
low). So, while I agree that regions truly known to be weakly coupled are interesting, regions
for which we have no real constraint are equally important targets for "better understanding
the slip behaviors during earthquake cycles". The 1938 zone is not 'weakly coupled', but
your own study shows how valuable it is to know the slip behavior for at least two large
events in the area.

Those are good points. We reworded this sentence to “Given the limited records, it is ~~still~~
often uncertain whether ~~the~~ more weakly-coupled “seismic gaps” ~~still~~ have the ability to
produce a great megathrust earthquake and generate large tsunami.”

line 108. The 2005 event was Mw 8.7; that is rather a poor comparison to make with the 7.8
in the 2020. Hard for such a small event to have ruptured to the trench. Yes, all models
suggest that it did not, but the better comparison would be with a 7.8 that ruptured depths of
20-40 km and also reached to the trench, and I cannot think of a case. So, the comparison is
not very informative.

Thanks, we decided to delete this sentence and associated ref. 28. It now reads “The
earthquake rupture is located in the eastern part of the Shumagin Gap, and the coseismic slip
concentrates between 30 km and 40 km depth beneath the Shumagin Islands with a peak slip
of 2.6 m, and it does not reach the trench”.

line 219. Given my scepticism about resolution, I have a hard time believing that one can
resolve the locking area to be only 25% of the coseismic slip area. Yes, the preferred models
do that, but is it not a consequence of some effort in the inversion to find a minimum area
intrinsically? I have no problem with the physical plausibility of this finding, but am dubious
that one has real resolution of pin-point locking. Not sure what to recommend other than
perhaps some discussion so that readers know what to believe versus what is an inversion
bias. My unease comes from comparing the 'kinematic' and 'stress-driven' afterslip
distributions seen in Figure 3; the highly parameterized 'physical' model gives results that are
very different in absolute slip placement from the kinematic solution. This shows that the
data can be fit by such different models that it is frustrating (that is just the way it is), and it
makes me doubt any claims of precise resolution of locking areas.

We characterize the preferred coupling model shown in Fig. 5 as “plausible” and provide
tests showing that substantially larger coupled asperities severely degrade the fit to the
interseismic velocities. We inserted the following sentence in Lines 495-497 (tracked PDF
file) “While details of the locking distribution are still not well resolved (see Supplementary
Figs. 13, 15-18), we can rule out scenarios with much larger locked zones, such as those
associated with inferred historical rupture areas (Supplementary Fig. 12)” .

line 340-341 the sentence here is confusing: and needs some clarification: "After subtracting
the moment released during the 2021 Chignik earthquake in this zone, which is
approximately 50% of the total moment of the 2021 event, the remaining cumulative moment
deficit is estimated to be 2.9×10^{21} Nm ($M_w=8.2$)."

We have rewritten this sentence to make it easy to follow. It now reads (now Line 513 of the
tracked PDF file): "After subtracting the moment released during the 2021 Chignik
earthquake in the overlapping region with the 1938 rupture zone, the remaining cumulative
moment deficit is estimated to be 2.9×10^{21} Nm ($M_w=8.2$)".

Overall, this is excellent work, and I encourage consideration of some of my issues, but only
minor revisions are needed for this to be an acceptable contribution in Nature
Communications. Thorne Lay.

Great thanks again!

Reviewer #3 (Remarks to the Author):

In their paper "Aseismic slip and recent ruptures of persistent asperities along the Alaska
Aleutian subduction zone" Zhao et al. present a co-seismic and early post-seismic slip model
(based on kinematic and stress driven modeling) for the 2020 M7.8 Simeonof earthquake.
They link their findings to slip from the recent 2021 M8.2 Chignik earthquake and other large
events in the Alaska-Aleutian subduction zone to infer a conceptual model of heterogenous
coupling distribution, and stable asperities in this region. They argue that the latter are stable.

This is an interesting paper that brings a different dimension to the existing, purely
kinematically derived interseismic coupling models for the region. The findings for the
afterslip distribution, both from kinematic and stress driven modelling are an important
update to the findings that exist so far (Crowell and Melgar, 2020), which were based on very
limited data. The data analysis and modeling methodologies are sound, the interpretations
generally robust. While I have a few questions and concerns that I outline below, I do think
the timeliness and broad impact of the paper warrant publication in Nature Communications
after minor revisions.

Thanks for your positive evaluation.

My main concern with the paper is the suggestion that these asperities are persistent. There's
ample evidence for significant variability in terms of rupture size in the general region (e.g.,
Shennan et al., 2009 & 2014; Briggs et al., 2014) and I don't quite understand how this
meshes with the proposed persistent asperity hypothesis. The main difference between the
existing literature (e.g., Li & Freymueller; Drooff & Freymueller) is that the current paper
proposes a rather heterogeneous along dip segmentation of coupling, based on historic and
recent events (and some trial and error modeling). They recognize that the along dip coupling
is difficult to impossible to constrain with the existing data. So, instead of having to justify a
regularization strategy that partitions the along dip coupling, they lean on locations of events.
The trouble with those is that they need to have happened first and unfortunately, the authors
cannot either provide much certainty on the characteristics of the shallow interface.
Remarkably, their results generally match the along-strike segmentation, which remains quite
stable in different approaches of kinematic coupling inversions using the GNSS data. This, I
find is the important finding of their analysis. In light of this, I would appreciate if the authors
would more prominently discuss the assumptions of their asperity model and what their
reasons are to assume long-term persistence of these.

We acknowledge that the existence of persistent asperities is still a hypothesis and remains
somewhat controversial. However, we think the question posed here focused more on the
“characteristic” failure of individual asperities, not necessarily the spatial distribution of
asperities. In this paper, we are focusing on the “persistent locked patches”; that is, the
patterns of interseismic coupling on the slab interface probably persist over time spans of one
or more earthquake cycles. The persistence of asperities does not necessarily produce
characteristic seismic ruptures (Avouac, 2015). The locked persistent (core) asperities
assumed in the paper are different from the final rupture size. As mentioned in the main text,
the size of the locked core asperity is approximately 25% of the coseismic model inferred
from inversion of geodetic and seismic datasets or of the rupture zone constrained from
aftershocks. We suggest that the core asperities are velocity weakening throughout the
earthquake cycle, and interseismic creep occurs outside of the asperities. However, once a
large earthquake nucleates in the unstable core asperities, it may propagate into surrounding,
conditionally stable areas and sometimes may break creeping segments that are otherwise
considered to be barriers to rupture propagation (Scholz, 1998; Noda & Lapusta, 2013).

Multiple asperity failures can occasionally happen in variable configurations. This is
probably why the rupture sizes and slip distributions are found to vary between different
earthquake sequences along the same segment (e.g., Park & Mori, 2007; Shennan et al., 2009,
2014; Briggs et al., 2014).

We included these points on the persistence of asperities in Lines 523-525 (tracked PDF file)
and Lines 528-529 (tracked PDF file), as shown below:

“In this paper, we focus on the spatial distribution of persistent locked asperities, that is we
assume that patterns of interseismic coupling on the plate interface persist over time spans of
one or more earthquake cycles”.

“However, the persistence of asperities does not necessarily produce characteristic seismic
ruptures⁵⁹, and in some cases, multiple asperities may rupture together⁶⁰.”

The assumptions of the asperities can be found in Lines 697-710 (tracked PDF file), as
shown below:

“We employ an “asperity model” to infer a physically plausible distribution of persistent
asperities. It is worth noting that the considered persistent asperities are based on the
inference of overlap of earthquake rupture zones and interseismic slip deficit zones. In this
simple model, the fully coupled patches are assumed to be confined to velocity-weakening
asperities, which are fully locked during the interseismic period, and postseismic and
interseismic creep occurs on the velocity strengthening area of the fault outside of the
asperities^{9,37}.”

Yes, you are right. This method relies on the knowledge of historic ruptures and tests the
assumption that these rupture asperities represent persistent locked patches. Fortunately, the
historic earthquakes were relatively well documented and observations span a substantial
fraction of likely recurrence intervals. As we have tested and shown in the supplementary
materials, we cannot assess the existence of asperities at shallower depths due to the lack of
GPS stations close to the trench. However, it seems that the better we know the size and
location of asperities located at depths of ~30-50 km, the more confidently we can infer the
state of coupling at shallower depths, since there are tradeoffs between asperities in the
shallow and deep depth intervals. Ultimately, seafloor geodetic data will reveal additional

details about the nature of the shallow portions of the megathrust. We documented these
points in Lines 489-494 (tracked PDF file), as shown below:

“We suggest that the better we know the size and location of asperities located at depths of
~30-50 km, the more confidently we can infer the state of coupling at shallower depths, since
there are tradeoffs between asperities in the shallow and deeper depth intervals. Ultimately,
seafloor geodetic observations in this and other partially coupled subduction zones will help
534 us to better constrain the size and location of seismic asperities and to assess the associated
tsunami earthquake hazard”.

The other major comment I have is regarding the stress-driven afterslip model. Yes, the
authors are correct that this is more physical, but the fit to the data, particularly in the near
field and down-dip their rupture zone (AC28, AV07, AC25, AC21) is not good. This can
have several reasons. One - this model is driven by the co-seismic slip distribution which is
derived solely from GNSS data, but those are relatively sparse in the area. So this postseismic
misfit could represent issues with the co-seismic model. Two - assumptions made for the
parameters governing the postseismic slip are not good. The fact that the model overpredicts
uplift at AC28 and subsidence at AB07 signifies that the amount of afterslip in the downdip
region is too large (it is not clear to me whether the afterslip is purely thrust, too, or has a
strike slip component to it). Since the authors lean heavily on this model throughout the
paper, I would request they either optimize the fit, or provide a more detailed discussion on
why these misfits are acceptable. The current mention in L160-161 that "the position time
series from this afterslip model are also in good agreement with the observations (Fig. 4)." is
a bit too optimistic. Furthermore, it would be instructive to explain why the stress driven
model suggests such localized, high amplitude afterslip, whereas the kinematic data-driven
inversion requires much more distributed, but lower amplitude slip.

Thanks for your comments on the stress-driven afterslip models. You are right. The resulting
stress-driven afterslip models heavily depend on the used coseismic slip models, which has
been documented by Johnson et al. (2006) and Barbot et al. (2009). We tested whether the
frictional afterslip models driven by other coseismic slip models, which were constrained by
different datasets, can improve the fit between the observed and modeled postseismic GPS
time series (Please refer to the replies to reviewer #1 and Figs. R5-R7). The tested frictional
afterslip models, which are driven by the coseismic models of Liu et al. (2020), Xiao et al.
(2021) and our coseismic model tests with different rake constraints, are shown in Fig. R4c.

They all produce slightly larger misfit values ($4.2 < \text{normalized } \chi^2 < 5.5$) compared to our
preferred model presented in the main text (normalized $\chi^2 = 3.0$). We have documented these
test results in Lines 207-209 (tracked PDF file) and added Figs. R5-R7 in the Supplementary
Materials (Supplementary Figs. 9-11).

Yes, we also noticed the substantial residuals in the vertical time series at AC28 and AB07.
As mentioned before, we find that it is not easy to reduce the misfit by using more complex
coseismic slip models, considering the rake variations, and adjusting the parameters
governing the rate-strengthening friction law. We could probably improve the fit to vertical
displacements by considering heterogeneity in frictional parameters as explored in Johnson et
al. (2006) for the afterslip following the 2004 Mw 6.0 Parkfield earthquake. However, we do
not think we can justify doing so given the small number of GPS sites. Therefore, we have
added a sentence in Lines 210-213 (tracked PDF file) to explain this. It now reads “*We could*
*probably improve the fit to the data by considering spatial variations in frictional parameters*
*as explored in Johnson et al.³¹ for the afterslip following the 2004 Mw 6.0 Parkfield*
*earthquake. However, we do not think we can justify doing so given the small number of GPS*
*sites”.*

We have revised the sentence in L160-161 (Line 188 of the tracked PDF file) as follows:
“*The position time series from this afterslip model are generally in agreement with the*
*observations (Fig. 4)”.*

As for the contrast in afterslip magnitude, in some previous studies, the slip of the kinematic
model is quite similar to that derived from the stress-driven afterslip model. For example,
Wang & Bürgmann (2020) obtained a similar distribution and magnitude of afterslip in their
kinematic and stress-driven afterslip models following the 2017 Mw 7.3 Sarpol-e Zahab, Iran
earthquake. In other cases, the magnitude of stress-driven afterslip, especially in the near-
field of the rupture, is much larger than that obtained in kinematic inversions (e.g., Diao et
al., 2020). We think the spatial sampling of geodetic data used in the kinematic inversion is
the key factor determining the inverted slip magnitude. The choice of smoothing parameters,
of course, is another reason. In our study, only few GPS stations are available to invert for
afterslip. Therefore, the inverted afterslip region is widely distributed and the resulting peak
slip is relatively small, even though the moment released in the areas surrounding the rupture
is comparable. We have explained the reason in the main text Line 383-385 (tracked PDF
file), as follows “*Compared to the stress-driven afterslip model in Fig. 3b, the distributed and*

small magnitude of kinematic afterslip (Fig. 3a) is mostly due to the small number of GPS
stations used in the kinematic inversion".

Not as important as these first two points, but worth mentioning before the minor comments
is the statement in L289 where a relationship between afterslip and aftershocks is drawn.
"This suggests that the aftershocks in the first 87 days may be mainly triggered by aseismic
afterslip in the peripheral region of the coseismic area." This seems to jump the gun and
mistake (temporal) correlation for causation. Wouldn't you want a better spatial correlation
between afterslip maxima and aftershock locations? Figure 6A suggests very poor spatial
correlation. If this were true, I would expect most of the aftershocks in the high afterslip
regions, but we don't see this. Based on figure 3F one could make the argument that most of
the aftershocks occur in a gap of downdip and updip afterslip to the west of the rupture zone,
releasing the co-seismically generated stresses seismically, rather than aseismic. The same
may be roughly true for the updip cap, although that one cluster of aftershocks locates a bit
too far west for that.

Thanks for your comments. We have revised this part according to your suggestion. See
Lines 427-429 (tracked PDF file): "The good temporal but poor spatial correlation between
afterslip and aftershocks probably reflects that many aftershocks were triggered by static and
dynamic stress changes from the event, not just the failure of small asperities driven by
surrounding afterslip".

I consider these comments minor, as I don't expect that the major points of this paper will be
significantly altered, nor should it take too long to address these points. I hope to see this
published in Nature Communications in the near future!

Minor comments:

L32 "earthquake" delete "s"

Done.

L35 "normal earthquakes" seems ambiguous in the context "regular"?

We have revised this to “regular earthquakes”.

L248: Invoking a slow slip event here seems maybe a step too far. What is the evidence for
this? And how does afterslip differ from a slow slip event in this case?

We consider afterslip to be postseismic slip that relieves coseismic stress changes, while a
triggered slow slip event may relieve a much more substantial slip deficit that has built up
with time (e.g., see definitions in Table 1 and related text in Bürgmann, 2018). Thus, the
evidence for the occurrence of a triggered SSE comes from inverted slip that is greatly in
excess of that predicted from the stress-driven afterslip model. We have clarified this as
follows, now in Lines 376-379 (tracked PDF file): “. *The frictional afterslip models produce*
*less slip to the west of the rupture than found in the kinematic inversion. That is, the*
*coseismic stress change cannot be solely responsible for the observed slow slip in that area.*
*This may suggest the occurrence of a triggered slow slip event, which released a previously*
*built-up slip deficit on that portion of the Shumagin Gap³⁸*”.

L267 "We suspect that the poor correlation between the spatial distribution of the aftershocks
and afterslip is in part caused by large uncertainties in the aftershock locations" This seems at
odds with the comment about a good spatial correlation between the two just lines before this.
Could you at least give an estimate of the location error?

We have rewritten the statement for consistency with the statement in L267. New Lines 427-
429 (tracked PDF file): “. *The good temporal but poor spatial correlation between afterslip*
*and aftershocks probably reflects that many aftershocks were triggered by static and dynamic*
*stress changes from the event, not just the failure of small asperities driven by surrounding*
*afterslip*”.

We do not know the aftershock location errors in this region, but teleseismic event locations
can be off by 10s of km due to Earth structure effects. We have added a sentence in Line 407
(tracked PDF file), “. *since teleseismic event locations can be off by 10s of km due to Earth*
*structure effects⁴⁷*”.

L293: "The seismic moment released by the aftershocks in the 87-day postseismic period is

about two orders of magnitude smaller than the moment released by aseismic afterslip
(Supplementary Fig. 14)." What does this tell us? This statement isn't picked up anywhere.

We have added a sentence to tell the readers what we conclude from the comparison between
the seismic moment and aseismic moment after the main shock; Line 434 (tracked PDF file)
"suggesting that the coseismically increased stress surrounding the rupture is mainly relieved
through aseismic slip in the first 87 days."

L327 Is the fit really equally good?

Strictly speaking, the fit between our observations and predictions from the preferred
interseismic asperity model (WRMS=1.9 mm and normalized $\chi^2=13$) is somewhat worse
than the previously published kinematic models, for example, in Li & Freymueller (2018)
(WRMS=1.1 mm and normalized $\chi^2=4.1$). As mentioned in the text (Lines 265-273 of the
tracked PDF file), the poor fit in the displacement azimuths of GPS stations above the 1938
and 1964 rupture zones is probably mainly attributable to the unmodeled complex forearc
deformation, as well as other reasons mentioned in the *Results* section. Thus, we have
rewritten this sentence in Lines 474-484 (tracked PDF file) as "Our more physical
interseismic asperity model, assuming full locking on core rupture asperities of historical and
more recent earthquakes, shows possible but not unique backslip scenarios that fit the GPS
measurements generally well, except in the areas above the 1938 and 1964 rupture zones as
described before (Supplementary Fig. 14). Our preferred but non-optimized forward model in
Fig. 5 produces a WRMS value of 1.9 mm/yr that is slightly larger than 1.1 mm/yr for the
inverted backward slip model in Li & Freymueller²² (see Methods for details on the GPS
velocities used in misfit calculation). It is not surprising that they obtained lower residuals,
since they also estimated the strike-slip component on each fault patch (Supplementary Fig.
14)."

L331 makes the same argument that the existing kinematic coupling models make - there's no
data near the trench, so we don't know. The difference is that the coupling in the Li &
Freymueller, and Drooff & Freymueller models is uniformly larger, rather than asperity
driven, which we only know about after the fact.

Thanks for your comments. We believe that the following wording further clarifies this
situation. We now state that "We suggest that the better we know the size and location of

*asperities located at depths of ~30-50 km, the more confidently we can infer the state of*
*coupling at shallower depths, since there are tradeoffs between asperities in the shallow and*
*deeper depth intervals.*” after L331 (Now Lines 489-492 of the tracked PDF file).

L339 - What is the "red curve" in figure 5?

We have modified this to make it clear. It now reads “*the moment deficit over the 1938*
*rupture zone bounded by the dark red curve in Fig. 5*”.

L422 - Why no strike slip component in the slip model? Other models have shown moderate
(Liu et al, 2021) to significant (Ye et al, 2021) along strike motion.

We did not consider the possible strike-slip component in the coseismic and afterslip models
because the focal mechanisms from the GCMT indicate the earthquakes around the slab
interface were pure thrust faulting events. We have reinverted for coseismic slip models with
different rake constraints and different regularization methods and find that the obtained
coseismic slip models with rake allowed to vary between 45~135 deg are similar to those
with rake fixed at 90 deg (Fig. R4). In addition, we have tested whether we can improve the
fit to the GPS-observed postseismic time series, especially in the vertical component, when
invoking rake variation in the coseismic slip models driving the stress-driven afterslip
models. However, we find that the fit to the GPS data cannot be improved when considering
source models with a strike slip component. **For more details and related changes in the**
**manuscript please refer to our responses to Reviewer #1 and Fig. R3-R7.**

L465 - specifically, what are the burn in and thinning parameters applied? What's the auto
correlation of the remaining samples?

The Monte-Carlo results have been obtained using the emcee package
(<https://github.com/dfm/emcee>), with a burn value of 20000 and default thinning value of 1.
The autocorrelation of the Markov chain decreases quickly with an increasing lag and
becomes virtually zero within a lag of ~100-200 (Fig. R11). These results show that the
chains are sufficiently converged. We have added these details in Lines 666-669 (tracked
PDF file) in the Methods section as shown below:

“Our Monte Carlo chain has 40,000 samples and produces 20,000 samples of the posterior
distribution with a default thinning value of 1. The autocorrelation of the Markov chain
decreases quickly with an increasing lag and becomes virtually zero within a lag of ~100-200
(Supplementary Fig. 23)”.

We added Fig. R11 (now Supplementary Fig. 23) in the supplementary materials.

Figure R11. The normalized autocorrelation function of the stochastic process that generated the
chains for frictional parameter and reference velocity. Results for three scenarios of downdip-
only, updip-only and fully surrounding the coseismic slip zone are shown in **a**, **b** and **c**,
respectively.

References:

Avouac, J.-P., 2015. From Geodetic Imaging of Seismic and Aseismic Fault Slip to Dynamic
Modeling of the Seismic Cycle. *Annu. Rev. Earth Planet. Sci.* 43, 150223150959000.
<https://doi.org/10.1146/annurev-earth-060614-105302>

Barbot, S., Fialko, Y., Bock, Y., 2009. Postseismic deformation due to the Mw 6.0 2004
Parkfield earthquake: Stress-driven creep on a fault with spatially variable rate-and-state
friction parameters. *J. Geophys. Res.* 114, 1–26. <https://doi.org/10.1029/2008JB005748>

Briggs, R.W., Engelhart, S.E., 2014. Uplift and Subsidence Reveal a Nonpersistent
Megathrust Rupture Boundary (Sitkinak Island, Alaska). *Geophys. Res. Lett.* 41.

Bürgmann, R., 2018. The geophysics, geology and mechanics of slow fault slip. *Earth Planet.*
*Sci. Lett.* 495, 112–134. <https://doi.org/10.1016/j.epsl.2018.04.062>

Diao, F., Wang, R., Xiong, X., Liu, C., 2021. Overlapped postseismic deformation caused by
afterslip and viscoelastic relaxation following the 2015 Mw 7.8 Gorkha (Nepal)
earthquake. *J. Geophys. Res. Solid Earth.* <https://doi.org/10.1029/2020JB020378>

Johnson, K.M., Bürgmann, R., Larson, K., 2006. Factional properties on the San Andreas
fault near Parkfield, California, inferred from models of afterslip following the 2004
earthquake. *Bull. Seismol. Soc. Am.* 96, 321–338. <https://doi.org/10.1785/0120050808>

Nikkhoo, M., Walter, T.R., 2015. Triangular dislocation: an analytical, artefact-free solution.
*Geophys. J. Int.* 201, 1119–1141. <https://doi.org/10.1093/gji/ggv035>

Park, S.-C., Mori, J., 2007. Are asperity patterns persistent? Implication from large
earthquakes in Papua New Guinea. *J. Geophys. Res.* 112, B03303.
<https://doi.org/10.1029/2006JB004481>

Scholz, C.H., 1998. Earthquakes and friction laws. *Nature* 391, 37–42.

Shennan, I., Barlow, N., Carver, G., Davies, F., Garrett, E., Hocking, E., 2014. Great
tsunamigenic earthquakes during the past 1000 yr on the Alaska megathrust. *Geology*
687–690. <https://doi.org/10.1130/G35797.1>

- Wang, K., Bürgmann, R., 2020. Probing Fault Frictional Properties During Afterslip Updip
and Downdip of the 2017 Mw 7.3 Sarpol-e Zahab Earthquake With Space Geodesy. *J.*
*Geophys. Res. Solid Earth* 125, 1–22. <https://doi.org/10.1029/2020jb020319>
- Xiao, Z., Freymueller, J.T., Grapenthin, R., Elliott, J.L., Drooff, C., Fusso, L., 2021. The
deep Shumagin gap filled: Kinematic rupture model and slip budget analysis of the 2020
Mw 7.8 Simeonof earthquake constrained by GNSS, global seismic waveforms, and
floating InSAR. *Earth Planet. Sci. Lett.* 576, 117241.
<https://doi.org/10.1016/j.epsl.2021.117241>

REVIEWERS' COMMENTS

Reviewer #1 (Remarks to the Author):

The authors have thoroughly revised the manuscript and answered review comments sufficiently. It is an excellent contribution to Nature Communication and worth rapid publication.

Lingling Ye

Reviewer #3 (Remarks to the Author):

Ronni Grapenthin, UAF

I want to thank the authors for the excellent job they have done addressing all of my concerns. The authors did an excellent job addressing my concerns. I very much like the added materials in the supplements and recommend the paper for publication in Nature Communications.

I want to add an observation. When we worked on the Simeonof co-seismic slip model, we also wondered about the existing kinematic coupling models and their requirement of full coupling at the trench. In Xiao et al, 2021, we ended up relaxing this and instead regularizing this differently. The motivation was the question about why the Simeonof event would rupture in the less coupled part of that segment. The result, shown in Figure 7 of Xiao et al., is remarkable in two ways - the coupling in "Segment 1" in which the Simeonof event ruptured concentrated down-dip, closer to the Simeonof rupture (not perfectly aligned); and the coupling in the Western part of the Semidi segment ended up focused on the region where the Chignik event ruptured shortly after (the alignment there between coupling and our soon to be published co-seismic model is remarkable).

This changed coupling model doesn't fully agree with the asperity model put forward here. The main reason is our poor discretization constrained by the available data. However, it may serve as an additional argument for the validity of the model discussed here as there's now much closer alignment of the two approaches in that region than existed before - strengthening the argument of this paper.

I'll leave it up to the authors whether they want to expand their discussion slightly, and I certainly don't require an additional review of the paper. The main point of my comment here is to emphasize my excitement about this alignment.

A last comment, quite minor, is that the first authors lists

Institute of Raster Prevention

as his second affiliation. Does he perhaps mean "Disaster Prevention" ?

**REVIEWERS' COMMENTS**

Reviewer #1 (Remarks to the Author):

The authors have thoroughly revised the manuscript and answered review comments
sufficiently. It is an excellent contribution to Nature Communication and worth rapid
publication.

Lingling Ye

Thank you again Lingling for your previous valuable input and comments.

Reviewer #3 (Remarks to the Author):

Ronni Grapenthin, UAF

I want to thank the authors for the excellent job they have done addressing all of my concerns.
The authors did an excellent job addressing my concerns. I very much like the added
materials in the supplements and recommend the paper for publication in Nature
Communications.

Thanks for your positive evaluation and also for the previous comments.

I want to add an observation. When we worked on the Simeonof co-seismic slip model, we
also wondered about the existing kinematic coupling models and their requirement of full
coupling at the trench. In Xiao et al, 2021, we ended up relaxing this and instead regularizing
this differently. The motivation was the question about why the Simeonof event would
rupture in the less coupled part of that segment. The result, shown in Figure 7 of Xiao et al.,
is remarkable in two ways - the coupling in "Segment 1" in which the Simeonof event
ruptured concentrated down-dip, closer to the Simeonof rupture (not perfectly aligned); and
the coupling in the Western part of the Semidi segment ended up focused on the region where
the Chignik event ruptured shortly after (the alignment there between coupling and our soon
to be published co-seismic model is remarkable).

This changed coupling model doesn't fully agree with the asperity model put forward here.

The main reason is our poor discretization constrained by the available data. However, it may
serve as an additional argument for the validity of the model discussed here as there's now
much closer alignment of the two approaches in that region than existed before -
strengthening the argument of this paper.

I'll leave it up to the authors whether they want to expand their discussion slightly, and I
certainly don't require an additional review of the paper. The main point of my comment here
is to emphasize my excitement about this alignment.

We appreciate these comments and agree that comparison of our preferred interseismic
asperity model with the updated coupling model of Xiao et al. (2021) will strengthen the
arguments of our paper. It now reads "Recently, Xiao et al.³⁰ developed a new interseismic
coupling model with the assumption that the inverted coupling follows a Gaussian
distribution, rather than a decaying function, with depth. In this model the peak slip deficit is
centered in the intermediate depth range instead of near the trench as found in previous
models^{22,24}. Although their new model does not fully agree with the asperity models presented
in this study, it supports the idea that the 2020 and 2021 events did not rupture poorly
coupled sections of the megathrust interface." in Lines 324-329.

A last comment, quite minor, is that the first authors lists

Institute of Raster Prevention

as his second affiliation. Does he perhaps mean "Disaster Prevention" ?

Thanks for your careful reading. We have revised this typo.